# Elevation-dependent pattern of net $CO_2$ uptake across China

Da Wei [1,2,4] ✉, Jing Tao[1,2,4], Zhuangzhuang Wang[1,2], Hui Zhao [1], Wei Zhao[1] & Xiaodan Wang[1,3] ✉

The elevation gradient has long been known to be vital in shaping the structure and function of terrestrial ecosystems, but little is known about the elevation-dependent pattern of net $CO_2$ uptake, denoted by net ecosystem productivity (NEP). Here, by analyzing data from 203 eddy covariance sites across China, we report a negative linear elevation-dependent pattern of NEP, collectively shaped by varying hydrothermal factors, nutrient supply, and ecosystem types. Furthermore, the NEP shows a higher temperature sensitivity in high-elevation environments (3000–5000 m) compared with the lower-elevation environments (<3000 m). Model ensemble and satellite-based observations consistently reveal more rapid relative changes in NEP in high-elevation environments during the last four decades. Machine learning also predicts a stronger relative increase in high-elevation environments, whereas less change is expected at lower elevations. We therefore conclude a varying elevation-dependent pattern of the NEP of terrestrial ecosystems in China, although there is significant uncertainty involved.

Terrestrial ecosystems play a fundamental role in counteracting increasing $CO_2$ concentrations in Earth's atmosphere[1,2]. The net uptake of $CO_2$ by terrestrial ecosystems shows a geographical pattern (denoted here by net ecosystem productivity, NEP), with a peak in NEP in subtropical and temperate forests[3]. Similar to latitude, the elevation gradient has long been considered to exert an important physical control on vegetation, which can be traced back to Alexander von Humboldt in the nineteenth century[4], or even earlier. Similar phenomena were also recorded by Andrew D. Hopkins, who observed a progressive delay in leafing out with increasing elevation[5].

The elevation gradient affects temperature, precipitation and the supply of nutrients, with colder temperatures, lower precipitation and a smaller supply of nutrients at higher elevations[6]. This elevation-dependent pattern in the supply of resources largely determines both the structure of ecosystems (e.g., the ecosystem type, biodiversity, and soil characteristics) and their function (e.g., photosynthesis, phenology, and microbial activity)[4–7]. Several studies have been conducted across Earth's high-elevation environments, e.g., the Alps[8], Andes[9], and Tibetan Plateau[10], suggesting varying elevation-dependent patterns of photosynthesis and respiration. The strong elevation-dependent pattern of resource supply may also affect the NEP, i.e., the net balance between photosynthesis and respiration, but we do not have a broad picture of Earth's elevation-dependent pattern of NEP due to a lack of instrumental measurements of $CO_2$ exchange like eddy covariance observations.

High-elevation environments, i.e., high mountains (defined as >3000 m above sea-level), hold a disproportionally high proportion of Earth's biodiversity[11], despite their colder, drier climate and stronger nutrient limitation[12]. However, the current warming of Earth's climate will affect the net uptake of $CO_2$ in high-elevation environments by relieving the limitation of temperature. More importantly, these regions will experience elevation-dependent warming—that is, high-elevation environments will experience more rapid changes in temperature than lower-elevation environments[6]. In addition, there is less

[1]State Key Laboratory of Mountain Hazards and Engineering Safety, Key Laboratory of Mountain Surface Processes and Ecological Regulation, Institute of Mountain Hazards and Environment, Chinese Academy of Sciences, Chengdu, China. [2]University of Chinese Academy of Sciences, Beijing, China. [3]Institute of Tibetan Plateau Research, Chinese Academy of Sciences, Beijing, China. [4]These authors contributed equally: Da Wei, Jing Tao. ✉ e-mail: weida@imde.ac.cn; wxd@imde.ac.cn

direct disturbance from humans in high-elevation environments, implying the responses of their $CO_2$ uptake to climate change may differ from lower-elevation environments. Taken together, these factors suggest that climate warming in high-elevation environments could cause a more robust change in the uptake of $CO_2$ than at lower elevations, although evidence is still lacking.

China is the third largest country on Earth, and the elevation of China ranges from sea level to 8844 m and includes a wide range of tropical, subtropical, temperate, boreal, and alpine ecosystems. Roughly 65% of China's land surface is covered by mountains[13]. These mountains are home to 22% of China's population, indicating much lower human pressures than at lower elevations[13]; plus, they also receive much stronger protection and restoration efforts. For example, most newly planted forests are located in mountain environments[14]—whereas non-mountainous regions are occupied by croplands, buildings and other artificial land surfaces. Most protected areas are also in mountainous regions, such as the newly built San-jiangyuan (>4000 m) and Giant Panda (1500–3000 m) National Parks[14]. Existing studies have highlighted that these mountains function as a vital net $CO_2$ sink[3,15] and eddy covariance observations have been widely established in China during the last two decades[16–18]. China's topography and its eddy covariance network therefore provide a unique opportunity to explore the elevation-dependent pattern of NEP under the influence of a changing climate and human activities.

Therefore, in this study, we collected observations from 203 eddy covariance sites across China to determine the elevation-dependent pattern of NEP and its changes under a warming climate and changing human activities. We refer to this dataset as EddyChina2023 (Fig. 1a; more details of the dataset can be seen in Text S1, Figs. S1–S3 and Table S1). We first describe the pattern of NEP with elevation, as well as the environmental drivers of the elevation-dependent pattern of NEP. We then compare the difference between the high-elevation environments (3000–5000 m) and their lower elevation counterparts (<3000 m) in terms of the temperature sensitivity. We also explore the variation in NEP along the elevation gradient during the past four decades, based on biogeochemical models and satellite observations. Finally, we predict how the elevation-dependent pattern of NEP will be affected under various climate scenarios using machine learning, the eddy covariance dataset, and Coupled Model Intercomparison Project Phase 6 (CMIP6) models. Taking China as an example, the objective of this study was to draw a broad picture of Earth's elevation-dependent patten of NEP and to determine whether there will be a more robust or different change at higher elevations compared with that at lower elevations.

## Results

### Elevation-dependent pattern of NEP and its drivers
We first explore the elevation-dependent pattern of NEP. It shows a linear negative elevation-dependent pattern attributable to both the peak $CO_2$ rate and the length of the growing season (Fig. 1b, c)—that is, a shorter growing season and lower peak $CO_2$ uptake at higher elevations. There is a clear elevation-dependent pattern wherein a 100-m increase in elevation causes an NEP loss of about 4 g C m$^{-2}$ yr$^{-1}$. In fact, both gross primary productivity (GPP) and ecosystem respiration (RE) are significantly and negatively dependent on elevation, similar to NEP. Moreover, GPP and RE are highly correlated with each other, and RE consumes 72% of the GPP, leaving a 28% carbon use efficiency across all terrestrial ecosystem types (Fig. S4).

We find, based on >200 eddy covariance towers, that NEP and its components follow a linear negative pattern with an increase in elevation. Correlation analysis and structural equation model (SEM) analyses suggest that NEP is better correlated with GPP (Fig. 2a), rather than RE, indicating the dominant role of photosynthesis in affecting NEP[19]. Correlation analysis supports a dominant role for temperature, precipitation and the reactive nitrogen (N) level in shaping the spatial

pattern of GPP and RE (Fig. S5). SEM analysis further validates the importance of temperature, precipitation, and reactive N level to the spatial pattern of NEP. It is also clear that the temperature, precipitation, and reactive N level consistently show a negative elevation-dependent pattern (Fig. 2b), which contributes to a similar linear negative elevation-dependent pattern of NEP. Climate-induced land-cover and land-use types along the elevation gradient may also contribute to the varying elevation-dependent pattern of NEP. For example, there is more grassland, but less forest and cropland, at higher elevations (Fig. S6), among which forest and cropland have much higher NEP than grassland (Fig. S3). To summarize, the varying hydrothermal factors, nutrient supply, and ecosystem types may have collectively shaped the negative elevation-dependent pattern of NEP.

### Temperature sensitivity difference between high- and lower elevation environments
Further analyses validate our expectation that high-elevation environments and thus colder ecosystems (3000–5000 m; mean annual temperature of −0.12 °C, ranging from −6.0 to 8.7 °C), such as the Tibetan Plateau, are more sensitive to climate warming than their lower-elevation counterparts (mean annual temperature of 10.9 °C, ranging from −4.4 to 25.0 °C). We are already aware of the higher temperature sensitivity of RE in colder areas[20], which was also validated in our analysis (Fig. 3; Fig. S7), with $Q_{10} = 2.19$ in the high-elevation environments and $Q_{10} = 1.92$ at lower elevations ($P < 0.01$). Like the RE, the temperature sensitivity of GPP shows an elevation-dependent pattern: the temperature sensitivity of GPP in high-elevation environments is higher than that at lower elevations ($P < 0.01$). The higher temperature sensitivity of GPP in high-elevation environments thus leads to a stronger temperature sensitivity of NEP ($P < 0.01$). The higher temperature sensitivity of GPP, RE, and NEP in high-elevation environments are also well captured by the models' ensemble (Fig. S8).

It is notable that the temperature sensitivity of GPP in high-elevation environments is higher than that at lower elevations, while the temperature sensitivity of RE is not that significant, highlighting the role of GPP in dominating the temporal sensitivity of NEP. In terms of RE, it is well known that the carbon-rich soils accumulated in cold climates will suffer decomposition. Regarding the GPP, it is possible that the extent of temperature restriction has contributed to the significant variance in its sensitivity to temperature. Specifically, it appears that environments at lower elevations are approximately 11 °C warmer than high-elevation ecosystems. A similar phenomenon has also been recorded in the Scottish mountains, where greater phenological sensitivity to temperature was found, especially for spring growth at high elevations[21]. Another reason for the lower temperature sensitivity in lower-elevation environments may be the strong human impact present there, such as plantation, harvesting, logging, and land use changes. These factors may have partially masked the role of temperature in relation to GPP. Additionally, in populated lower-elevation areas, the availability of photosynthetic active radiation for plant growth may also affect photosynthesis and thus the variation in GPP[22]. In contrast, high-elevation environments typically experience intense solar radiation due to the thin air[23]. Thus, our eddy covariance dataset, in conjunction with process-based models, consistently demonstrates that the NEP is more sensitive to temperature variations in high-elevation environments than at lower elevations. This may be attributed to difference in temperature constraints, human impacts, and photosynthetic active radiation.

### Productivity change rates during the past four decades
NEP may have experienced varying rates of change under the varying climatic conditions along the elevation gradient—that is, it is uncertain whether high-elevation environments have become more productive during the last four decades as a result of significant changes in

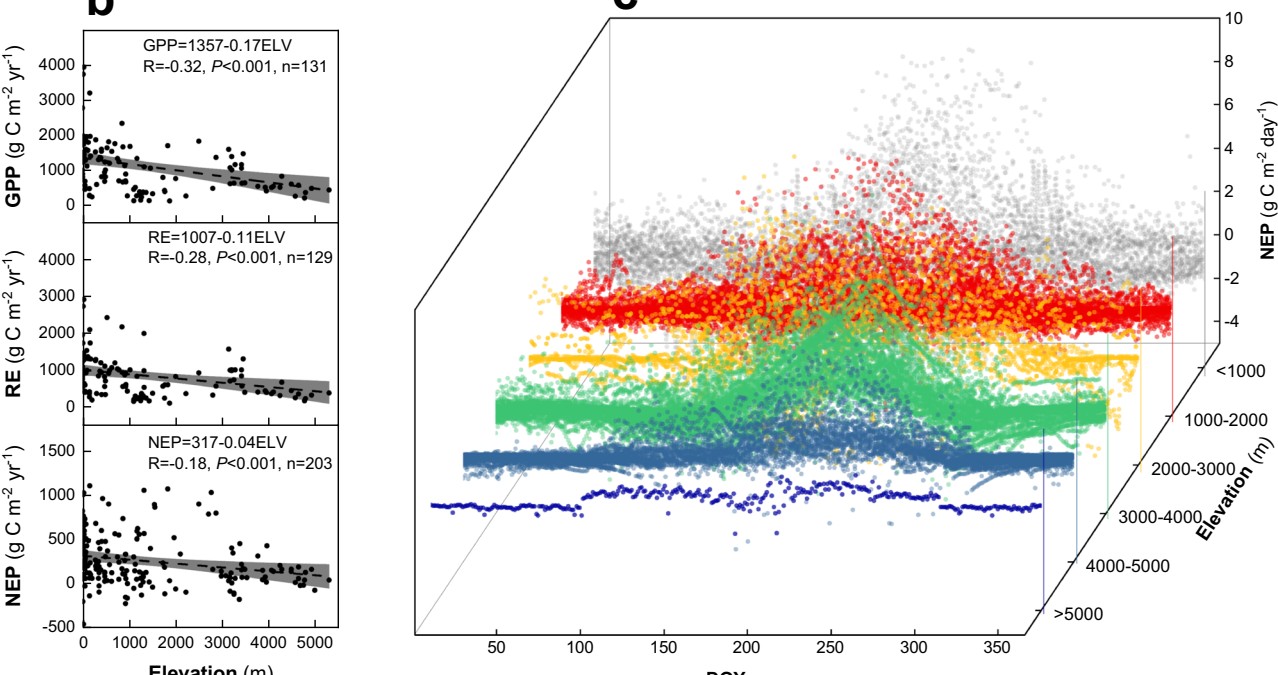

**Fig. 1 | Locations of eddy covariance sites and the variation of ecosystem characteristics in China. a** Locations of eddy covariance observation sites across China. To simplify the map, we only list the main ecosystem type for each station, though several towers may have been established to cover multiple ecosystem types for some stations (for details, see Table S1). **b** Variation in gross primary productivity (GPP), ecosystem respiration (RE) and net ecosystem productivity (NEP) along elevation gradients across China. The gray shading represents the 95% confidence band of the fits. **c** Temporal pattern of NEP at various elevation gradients across China. DOY=day of year. Source data are provided as a Source Data file.

precipitation, as well as changes in reactive N concentrations. It is notable that reactive N deposition has increased persistently across high-elevation environments in China[24]. This is different from lower elevations, where N deposition increased (1980–2000) and then

stabilized (after 2005), driven by socioeconomic changes and vigorous controls on N pollution[25,26]. We use the outputs of the Multi-scale Synthesis and Terrestrial Model Intercomparison Project (MsTMIP, Table S2)[27], which captures well the elevation-dependent pattern of

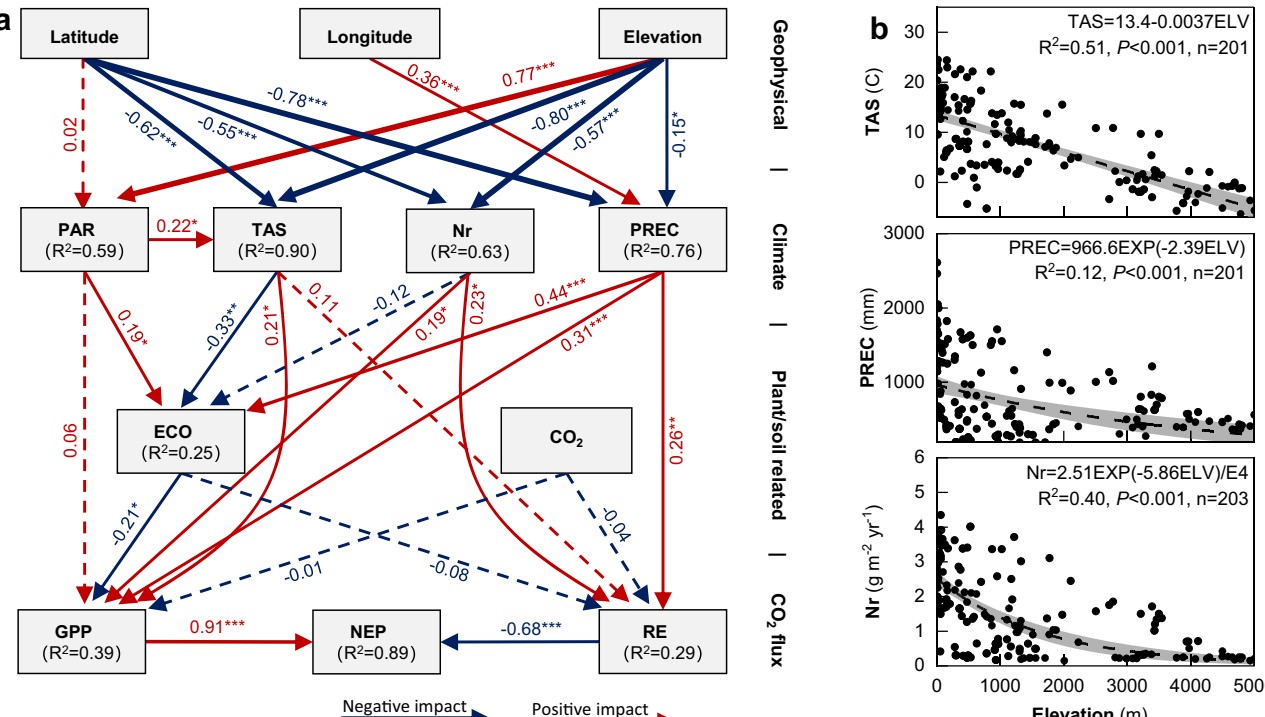

**Fig. 2 | Controls on the elevation-dependent pattern of NEP. a** Indirect and direct impacts of geographical factors, climate factors and soil processes on ecosystem $CO_2$ fluxes. GPP, gross primary productivity; RE, ecosystem respiration; NEP, net ecosystem productivity; PAR, photosynthetically active radiation; TAS, atmospheric temperature; Nr, reactive nitrogen; PREC, precipitation; ECO, Ecosytem type; $CO_2$, atmospheric $CO_2$ concentration. Given NEP represents the net balance between GPP and RE, we built its indirect relationship with environmental factors via GPP and RE. The structural equation model analysis was calculated based upon the annual average $CO_2$ fluxes (multi-year average used if there was more than one year of observations). The blue lines indicate negative impacts, while the red lines indicate positive impacts. The numbers adjacent to the arrows are the standardized path coefficients. The solid lines represent a significant correlation (*$P < 0.05$, **$P < 0.01$, ***$P < 0.001$), whereas the dashed lines indicate lack of significance. The $R^2$ values in the boxes indicate how much the dependent variables are explained by the independent variable(s). **b** Variation in TAS, PREC and Nr with elevation. The dashed lines indicate the linear or nonlinear fit for the environmental factors (the central estimate), whereas the gray shading represents the 95% confidence band of the fits. Source data are provided as a Source Data file.

NEP and GPP (Fig. 4a, b). Further analyses suggest that there is a consistent, more rapid relative increase in both NEP and GPP in high-elevation environments than at lower elevations, thereby validating our assumption that high-elevation environments are becoming more productive under global climate change.

Both the eddy covariance dataset and MsTMIP capture more robust changes in high-elevation environments, and this is also recorded by satellite-based GPP observations. The satellite-based GPP observations also capture the elevation-dependent pattern of average GPP (Fig. 4c), similar to the results based on both the EddyChina2023 dataset and the MsTMIP ensemble. The satellite-based GPP products further reproduce well the larger increase in GPP in the high-elevation environments, as compared with their lower elevation counterparts. Therefore, both the MsTMIP models' ensemble and satellite-based observations consistently reveal that high-elevation environments have become more productive during the last four decades. Higher temperature sensitivity may have contributed to the varying NEP rates of change in high-elevation environments by extending the growing season. For example, global warming has led to more uniform spring phenology across elevations in the Alps, i.e., a stronger phenological advance at higher elevations[5]. Also, stronger changes in precipitation in high-elevation environments may have contributed to their stronger rates of change in productivity, where the productivity of grasslands is strongly affected by precipitation[28,29].

### Projections of productivity change rates
It is uncertain whether future changes in climate, atmospheric $CO_2$, and reactive N will alter the elevation-dependent pattern of NEP (Figs.

S9–S12; Table S3). The eddy covariance dataset, CMIP6 models, and machine learning were therefore used to predict future variations in the elevation-dependent pattern of NEP under different climate scenarios (Fig. S13), including the Shared Socioeconomic Pathway (SSP) SSP1-2.6, SSP2-4.5, SSP3-7.0 and SSP5-8.5 scenarios. Future changes in climate and nutrient supply will cause a general increase in NEP across most elevation bands and the various climate scenarios (Fig. 5a–d). However, there are significant differences in the magnitude of their relative changes (i.e., the absolute changes divided by their mean value). Trying to validate the data-driven projections of NEP, we then compared our results with the default NEP outputs from CMIP6 (not constrained by EddyChina2023). Therefore, all scenarios, i.e., SSP1-2.6 to SSP5-8.5, consistently predict a stronger increase in NEP in the high-elevation environments (3000–5000 m), but less change in NEP at lower elevations (<3000 m), therefore affecting the slope of NEP along elevation under various climate scenarios (Fig. 5e–h).

The more robust relative change in NEP in the high-elevation environments than at lower elevations is generally consistent with the higher temperature sensitivity of colder regions. In addition, more robust changes in NEP in high-elevation environments can be attributed to changes in N deposition. There is only a slight decrease in N deposition in the high-elevation environments under SSP1-2.6 and SSP2-4.5, whereas there is an increase in N deposition under SSP3-7.0 and SSP5-8.5, which makes the more robust changes in NEP by plants possible. Recent evidence suggests that the high-elevation environments of China have experienced an increasing trend of reactive N deposition, which is different from the decreasing temporal trend from cropland/city sites at lower elevations[24]. Elevation-dependent

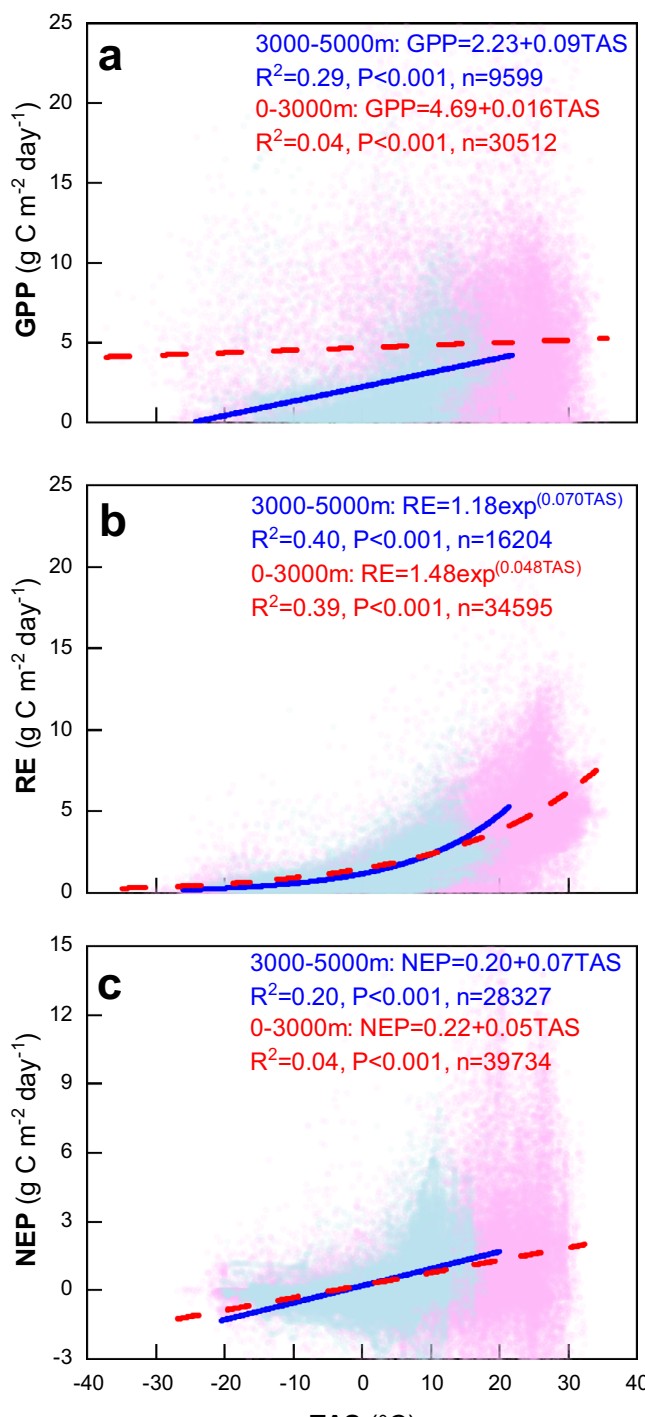

**Fig. 3 | Comparison of the temperature sensitivities of CO2 exchanges between high-elevation environments and their lower elevation counterparts. a** Response of gross primary productivity (GPP) to temperature variations in high-elevation environments and their lower elevation counterparts. **b** Response of ecosystem respiration (RE) to temperature variations in high-elevation environments and their lower elevation counterparts. **c** Response of net ecosystem productivity (NEP) to temperature variations in high-elevation environments and their lower elevation counterparts. General Linear Model univariate analyses were used to test the differences between the regression lines for GPP, RE and NEP. The dashed lines indicate the linear or nonlinear fits for GPP, RE and NEP. TAS, atmospheric temperature. The dashed line indicating the central estimate. Source data are provided as a Source Data file.

warming in high-elevation environments will also lead to a robust increase in mineralization[30], adding nutrients for plant growth and thus increasing NEP. We therefore predict a general increase in NEP under the various climate scenarios, with a more robust change in NEP in high-elevation environments, whereas lower elevations will be affected by future limitations in the supply of nutrients, despite increases in both $CO_2$ concentrations and temperature[31–33].

However, it is notable that the data-driven projections do have some uncertainty, as seen in the large uncertainty range in Fig. 5. The input states that statistical or "implicit" relationships were used to achieve data-driven projections between NEP and environmental factors, but it may have overlooked underlying mechanisms. Nonlinear changes and interactive effects from $CO_2$ concentration, climate extremes, species composition, and human management could also be difficult to capture through data-driven projections. Additionally, data-driven projections tend to focus more on aboveground factors like climate and vegetation, while neglecting drastic changes in underground processes, particularly in microbial processes, which may not be accurately reflected in eddy covariance observation or data-driven projections. Furthermore, projections based on data rely heavily on the availability of data, which is not always evenly distributed across different environments. For instance, there may be more observations in lower-elevation environments and fewer in high-elevation ones, which could introduce some degree of bias.

## Discussion

The elevation gradient is a vital driving force in shaping NEP in terrestrial ecosystems. The elevation gradient affects the climate, soils, vegetation, and nutrient supply, which have an important impact on NEP. Previous elevation-dependent studies have mainly focused on climate and biodiversity. Some studies have focused on alpine biomes, like the Alps[8] and Tibetan Plateau[10], but the broad picture of NEP along the elevation gradient is yet unclear. Taking the mountainous China as an example, we used an eddy covariance network to investigate the variations in NEP at different elevations. We found a linear negative elevation-dependent pattern in NEP, which is consistent across GPP and RE. This pattern was then validated by both model simulations and satellite observations. The elevation-dependent pattern of NEP is controlled by climate factors (temperature and precipitation), ecosystem type, and human activities (e.g., the supply of nutrients, such as reactive N). The eddy covariance dataset and models both supported well our expectation that higher and colder ecosystems may be more sensitive to climate warming, indicating an intrinsic feature in these high-elevation environments. The models and satellite observations showed that high-elevation environments respond consistently to a warming climate and may play a positive role in mitigating the effects of climate change.

Despite the scientific contributions mentioned above, our research has some limitations: (1) Our study was conducted at a national scale. Some unidentified factors may also affect the apparent $CO_2$ exchange recorded by the eddy covariance towers, such as the forest type/species composition, and the management, observation, and calculation procedures employed, but their contributions are currently difficult to determine. (2) Accurate climate data are important for analyses like those carried out in the present study. The climate data employed here for the temperature sensitivity analysis and Random Forest training were obtained from gridded climate datasets rather than ground-based datasets from each eddy covariance tower. This may have inevitably caused some uncertainty, despite these datasets being able to represent the elevation-dependent variation in both temperature and precipitation. To facilitate future analyses of the eddy covariance dataset, a data-sharing culture in the earth sciences

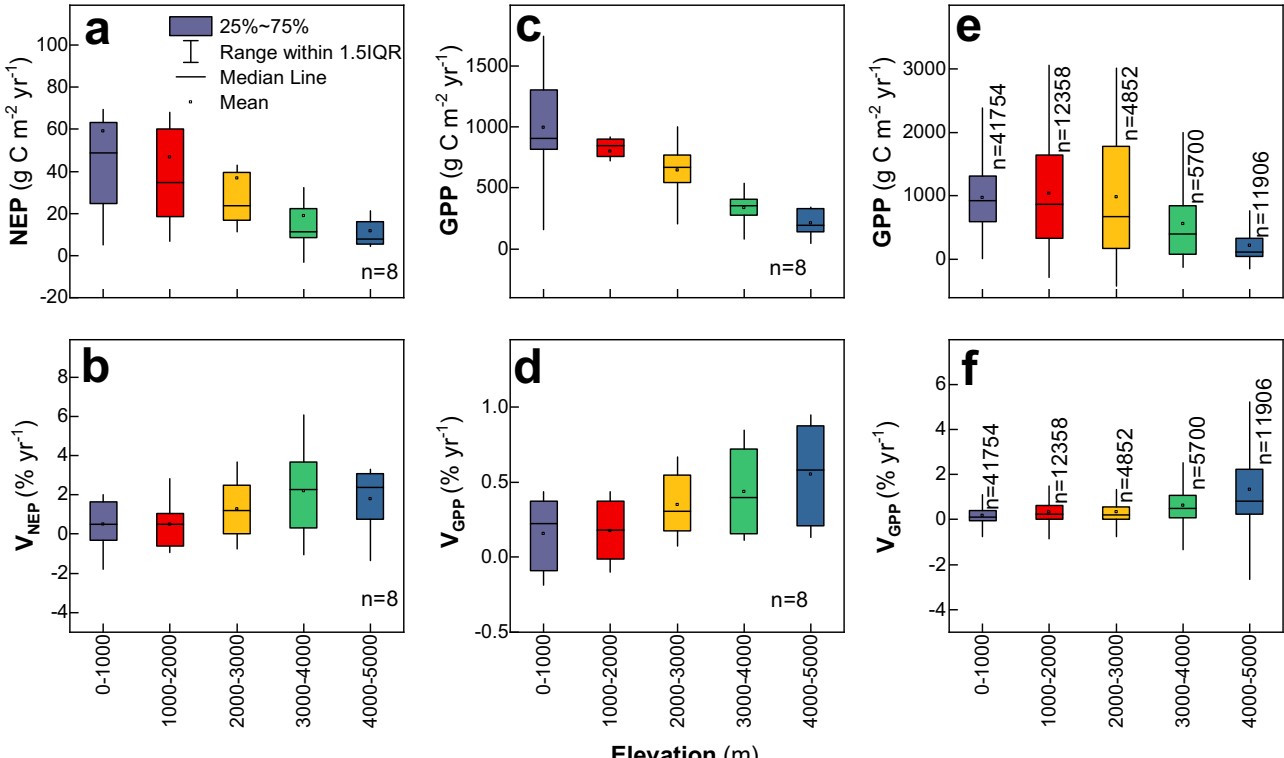

**Fig. 4 | Elevation-dependent pattern of net ecosystem productivity, gross primary productivity and their changes during the last four decades (1982-2018).** NEP, net ecosystem productivity; GPP, gross primary productivity. **a**, **b** Elevation-dependent pattern of the annual cumulative NEP and relative changes in the NEP with the MsTMIP model ensemble[27]. **c**, **d** Elevation-dependent pattern of the annual cumulative GPP and relative changes in the GPP with the MsTMIP model ensemble. The line extending from each bar represents the uncertainty in each climate scenario (±1 SD) of the MsTMIP model group (eight models: BIOME-BGC, CLASS-CTEM-N, CLM4, CLM4VIC, DLEM, ISAM, TEM6 and TRIPLEX-GHG). **e**, **f** Elevation-dependent pattern of GPP and relative changes from the GLASS GPP

product[37]. For the MsTMIP ensemble, all pixels of each elevation band were employed to derive the annual cumulative NEP or GPP for each model. Then, the results of these models were averaged to obtain the ensemble average NEP or GPP and the standard deviation for each elevation band. For the $V_{NEP}$, it was used to represent the speed of variation in NEP along the elevation gradient. The $V_{NEP}$ was calculated by dividing the slope of NEP changes by the average NEP: $V_{NEP}=Slope(NEP)/NEP_{baseline}$. In this equation, $NEP_{baseline}$ represents the annual average NEP during 1982–1985 (the first five years). The $V_{GPP}$ is calculated followed the very same procedure of the $V_{NEP}$. Source data are provided as a Source Data file.

should be encouraged in China[34]. (3) The future variation in NEP at lower elevations shows strong uncertainties, largely due to the uncertainties in the carbon capacity of forests and the supply of nutrients. China's afforestation campaigns during recent decades have transformed lower-elevation regions into hotspots of greening[19], although the future carbon storage capacity of forests is uncertain. Furthermore, the ongoing control of both pollutants and carbon emissions in China will reduce the supply of nutrients, though future variation in the level of reactive N is highly uncertain. (4) In high-elevation environments, there are large uncertainties regarding the vast areas of permafrost, which may provide positive feedback to the climate by releasing old carbon to the atmosphere[35]. However, there is scarce observational evidence relating to the vast area of permafrost in the interior of the Tibetan Plateau[15].

## Methods

### EddyChina2023 dataset
A total of 203 eddy covariance observation sites covering all ecosystem types and climate zones in China were identified (Table S1). Data were obtained for a total of 523 site-years (i.e., 2.5 years per site). Most of the data were obtained from peer-reviewed papers from the Web of Science (for English-language papers; http://apps.webofknowledge.com/) and the China National Knowledge Infrastructure (for Chinese papers; https://www.cnki.net/). The following keywords were used during the literature search: eddy covariance; $CO_2$ flux/exchange/sink/source; and China. Several criteria were used to guarantee the quality of the data in

the studies identified in the literature review. First, the data collection must have used the eddy covariance technique to ensure the spatial and temporal coverage. Observations based on static chambers, especially manual static chambers, were excluded because of their relatively small spatial coverage (typically <1 m²) and low temporal coverage (usually conducted in the daytime at weekly intervals). Second, the observations had to cover at least a whole year because $CO_2$ emissions outside the growing season significantly affect the annual-scale $CO_2$ sink. Third, the paper had to include a clear description of the eddy covariance installation, data collection and processing procedures (e.g., the Webb–Pearman–Leuning correction and axis rotation). All the sites were subjected to Webb–Pearman–Leuning density correction and were determined by the coordinate axis rotation approach—that is, 2D, 3D or planar fit coordinate axis rotation. The observational data had to be clearly and correctly presented, and papers with incorrect units of measurement were not included.

After applying these standards, the resultant EddyChina2023 dataset covered all the major ecosystem types of China−forest (61 sites), grassland (43 sites), wetland (38 sites), cropland (41 sites), desert (12 sites), and shrubland (10 sites)−with 203 sites in total. Among the data, 118 sites (58% of the dataset) covered 2+ years of observations, while 79 sites (40%) covered 3+ years. For each ecosystem type, 10+ sites were built to cover the NEP strength. In China, croplands, forests, and grasslands cover 3/4 of the land area. It is also notable that there were many eddy covariance observations in croplands (116 site-years), forests (134 site-years) and grasslands (129 site-

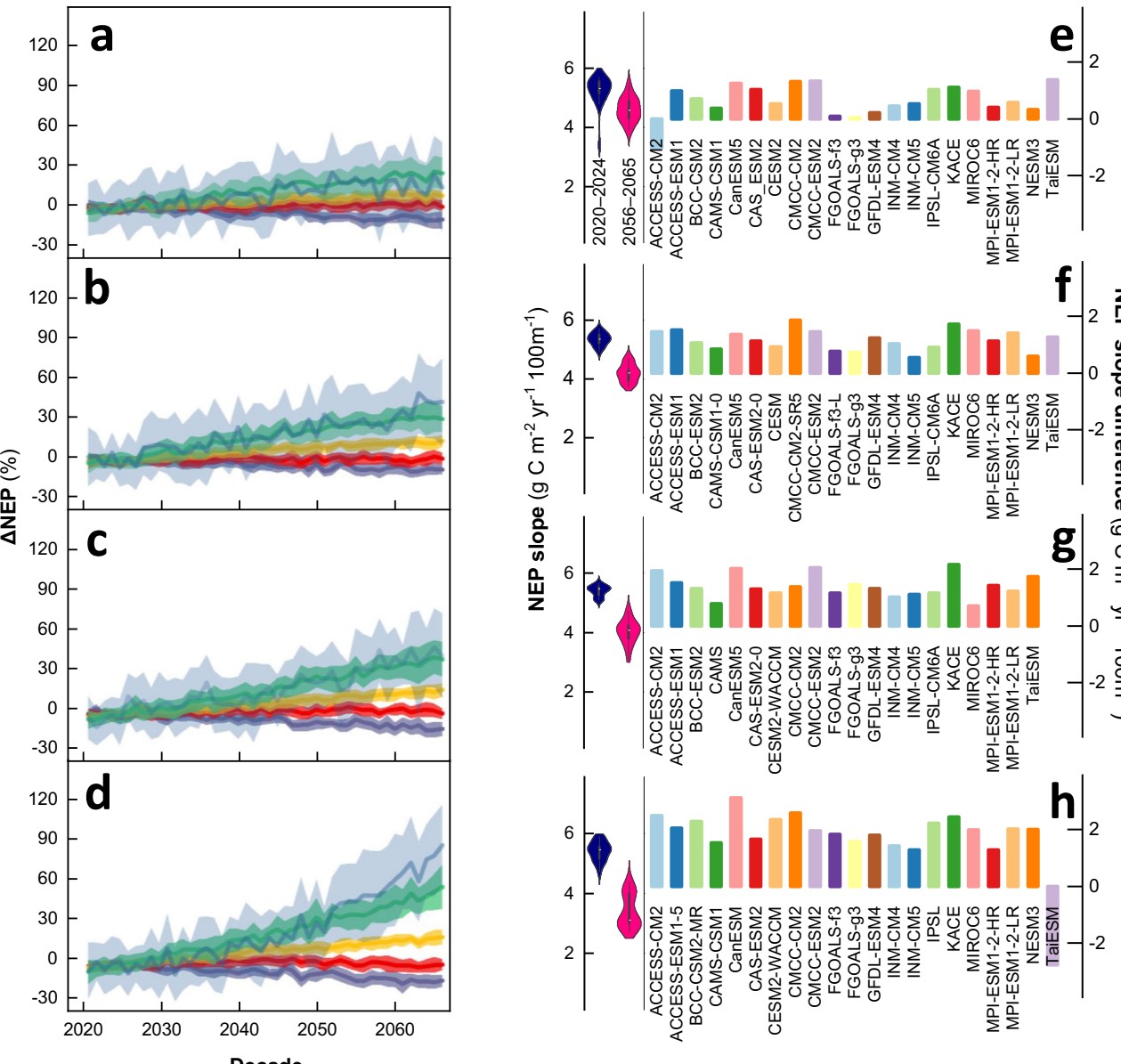

**Fig. 5 | Relative changes in net ecosystem productivity (ΔNEP) and NEP slope along elevation under various climate scenarios.** Relative changes in NEP under (**a–d**) SSP1-2.6, SSP2-4.5, SSP3-7.0 and SSP5-8.5. Relative changes were calculated by dividing the absolute changes in NEP by their average for 2020–2024: $\Delta NEP_i = (NEP_i\text{-}NEP_{baseline})/NEP_{baseline}$. Here, $NEP_i$ represents the NEP in year i, while $NEP_{baseline}$ represents the averaged NEP during 2020–2024. During the calculation, all pixels of each elevation band were employed to derive the annual cumulative NEP for each model under each climate scenario. Then, the results of these models were averaged to obtain the ensemble average ΔNEP and the standard deviation for

each elevation band under each climate scenario. Solid curves are the ensemble mean of the model simulations and the shading represents ±1 SD. Slope of NEP along elevation and its changes under various climate scenarios (**e–h**). For the left part of each figure, the NEP slope ($SL_{NEP}$) represents the slope between NEP with elevation. The bar represents the 1.5 times interquartile range. For the right part of each figure, the NEP slope change was the difference between 2020–2024 and 2056–2065: $Slope_{NEP}$ Difference = $Slope_{NEP\text{-}2020\text{-}2024}$ - $Slope_{NEP\text{-}2056\text{-}2065}$. Source data are provided as a Source Data file.

years). Part of the dataset was retrieved from ChinaFLUX and another literature-based dataset[15]. For those sites whose $CO_2$ fluxes could not be directly accessed, they were processed using GetData Graph Digitizer 2.2.6 (www.getdata-graph-digitizer.com). From the perspective of climate coverage, the eddy covariance sites were well distributed over China's climate gradient, albeit with relatively fewer sites in cold–dry and hot–wet areas, which is consistent with the limited number of sites in northwestern China. From the perspective of the study period, most observations were conducted during 2002–2020, with a mean of 2012 (Fig. S1). The dataset presented here is therefore representative of China in terms of its ecosystems, climate, and regional contributions, albeit with a slight regional bias.

## Structural equation model analysis

SEM analysis was used to estimate the relative importance of the direct and indirect impacts of geographical factors, climate factors and soil processes on the spatial pattern of $CO_2$ fluxes. For continuous observations of more than one year, the multi-year average value was calculated for each site. All NEP data are displayed as mean ± 1 SE values unless stated otherwise. It was hypothesized that the geographical factors (altitude and latitude) would largely affect the climate factors (radiation, mean annual precipitation and temperature), whereas the climate factors would directly and indirectly regulate $CO_2$ exchange via the ecosystem types. The site-averaged $CO_2$ exchange, geographical factors, climate factors, and soil factors were used in the SEM analyses.

For the climate factors (radiation, precipitation and temperature), the data for the nearest pixel to each site were extracted from the China Meteorological Forcing Dataset (CMFD), a high-resolution (0.1° × 0.1°, 3-h time step) climatological dataset (a simple evaluation of the CMFD dataset can be seen in Fig. S14)[36]. All the SEM analyses were preformed using IBM SPSS Amos (SPSS Inc., Chicago, IL, USA).

## Temperature sensitivity

For GPP and NEP, their temperature sensitivities were defined as the slopes of their variations on air temperature with linear regressions. For RE, the Van't Hoff equation was used to calculate the temperature sensitivity. i.e., $y = ae^{bt}$ (1) and $Q_{10} = e^{10b}$ (2). During the calculation, the daily average aboveground air temperature at 2 m was obtained from the CMFD dataset, given most towers did not provide air temperature observations in the literature. We then used General Linear Model (GLM) univariate analysis to test the differences regarding the regression lines for GPP, RE and NEP. During this process, the GPP, RE and NEP were the dependent variables, the different elevation groups (higher or lower than 3000 m in elevation) were the independent variables, while air temperature was covariate. For GPP, over 40,000 observations were involved in the GLM analysis, while there were more than 50,000 observations for RE and more than 60,000 observations for NEP. Daily accumulated or averaged GPP, NEP and air temperature data were used during the GLM analyses. For RE, given exponential regression models were used to reflect its temperature sensitivities, we applied a log transformation to the RE data before the GLM analysis. Effects were considered as significant at $P < 0.05$ (difference in temperature sensitivities between elevation bands), while the null hypothesis was accepted when $P > 0.05$ (no difference in temperature sensitivities between elevation bands).

## MsTMIP and GPP datasets

The MsTMIP dataset is a formal multiscale synthesis with prescribed environmental and meteorological drivers shared among model teams. The simulations are standardized to facilitate comparison with other model results and observations through an integrated evaluation framework (Table S2)[27]. The MsTMIP simulations were classified into four groups (BG1, SG1, SG2 and SG3). The BG1 group was simulated by the time-varying climate, land use, deposition of $CO_2$, and the reactive N input (eight models). BG1 mimics the ongoing changing Earth, enabling it to be compared with EddyChina2023. Within BG1, eight models provided the NEP or net ecosystem exchange outputs, including BIOME, CLASS-CTEM-N, CLM4, CLM4-VIC, DLEM, ISAM, TEM6 and TRIPLEX-GHG. All the models were driven by Climate Research Unit–National Centers for Environmental Prediction (CRUNCEP) climate data and the same soil map, and were run at a resolution of 0.5° with monthly time steps. For the GPP, we also used the Global Land Surface Satellite (GLASS) product. The GLASS GPP products are generated using integrated satellite GPP time series and Moderate Resolution Imaging Spectroradiometer surface reflectance data or Advanced Very High-Resolution Radiometer surface reflectance data[37]. Compared to other GPP products, the GLASS GPP product has been shown to have higher quality and accuracy[38], which has facilitated its wide application in global and regional studies, e.g., to drive process-based models, to evaluate land surface models, and to investigate vegetation dynamics under the changing environment[37].

## Future climate scenarios and machine learning

The prediction (during 2015–2065) of the net $CO_2$ sink of China's terrestrial ecosystems was driven by CMIP6 model simulations under various climate scenarios (Table S3). Specifically, four scenarios were used: SSP1-2.6, SSP2-4.5, SSP3-7.0 and SSP5-8.5. SSP1-2.6 is a combination of low societal vulnerability and a low forcing level, with a substantial change in land use (e.g., increased global forest cover). SSP2-4.5 is a combination of intermediate societal vulnerability and an intermediate forcing level. SSP3-7.0 combines relatively high societal vulnerability and a relatively high forcing level with substantial changes in land use (decreased global forest cover). SSP5-8.5 combines high societal vulnerability and a high forcing level.

The atmospheric [$CO_2$] shows a consistent increase by 2060: 478 ppm under SSP1-2.6, 535 ppm under SSP2-4.5, 587 ppm under SSP3-3.7, and 627 ppm under SSP5-8.5. Correspondingly, temperatures in China would increase by 1.3–2.6 °C by 2060 relative to the baseline period (2016–2020; annual average). All climate scenarios predict a general increase in precipitation of 48.9–86.7 mm by 2060 (annual cumulative), although there is strong spatial heterogeneity. The deposition of reactive N is projected to peak before 2030 and then decrease, consistent with the recently observed stabilization of reactive N deposition in China[25]. Forest cover is expected to benefit from land management and climate change[39], consistent with China's plans for forest plantations to double in size by 2050[40]. Vegetation will experience persistent greening, albeit with the magnitude differing among the various scenarios: the LAI is projected to increase by 18.7–30.1% by 2060 relative to the baseline period.

The Random Forest algorithm was used to conduct machine learning of the $CO_2$ sink. During the training, NEP was set as the label data, whereas environmental factors, like TAS, PRE, RAD, LAI and NR, were used as the input data. Before training, items without values were dropped from the datasets. Over 70,000 items of NEP for >130 sites were used for machine learning and the remaining sites that were not used were omitted because the studies that covered them did not include daily NEP data, instead only providing an annual average NEP rate. After normalization, 90% of the data were used for training and the remaining 10% for validation. They were trained 500 times to obtain a stable performance of the model (to keep a balance between a lack of fitting and over-fitting), with the mean square error used for the evaluation of model performance. Several datasets were used to retrieve environmental factors and simulations, including air temperature, precipitation, photosynthetically active radiation, and LAI. During the prediction, all driving data–temperature, precipitation, photosynthetically active radiation, and LAI–were obtained from CMIP6[41]. All datasets were resampled to a resolution of 0.1° × 0.1° via bilinear interpolation to facilitate model simulations and analyses. Before predicting future variation in NEP with Random Forest, the results of training Random Forest based upon the EddyChina2023 dataset were compared with FLUXCOM[42], a global-scale upscaling product based upon FLUXNET2015, regarding its seasonal pattern and magnitude (Fig. S15). The NEP outputs after training Random Forest were also in line with previous upscaling and remote sensing studies in China[19,43].

## Reporting summary

Further information on research design is available in the Nature Portfolio Reporting Summary linked to this article.

# Data availability

The EddyChina2023 data generated in this study have been deposited in the figureshae database under accession code https://figshare.com/s/e1f7f9c13e547e422a71. The MsTMIP data are available at https://daac.ornl.gov; the CMFD data are available from the National Tibetan Plateau Data Center (http://data.tpdc.ac.cn); the CMIP6 climate data are from https://esgf-index1.ceda.ac.uk; the land-cover data are from https://modis.gsfc.nasa.gov; and the GLASS-GPP product is from http://www.geodata.cn/thematicView/GLASS.html. Source data are provided with this paper.

# Code availability

The code data generated in this study have been deposited in the figureshae database under accession code https://figshare.com/s/87b86f36ef064f0bc599.

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

## Acknowledgements

We appreciate the efforts of ChinaFLUX, Professor Jialiang Tang, Dr Ning Ma and many other scientists for sharing their eddy covariance data. We thank Dr Yuan Zhang for providing the forest age map of China. We also thank Yahui Qi, Jiabin Fan and Zhaoheng Deng for their help during the preparation and revision of the manuscript. This work was supported by the Second Tibetan Plateau Scientific Exploration (2019QZKK0404, X.D.W.), the National Natural Science Foundation of China (41971145, D.W.), the West Light Scholar of the Chinese Academy of Sciences (xbzg-zdsys-202202, D.W.), the Youth Innovation Promotion Association of the Chinese Academy of Sciences (2020369, D.W.), the Science and Technology Major Project of TAR (XZ202201ZD0005G04, X.D.W.), and

the Science and Technology Research Program of the Institute of Mountain Hazards and Environment, Chinese Academy of Sciences (IMHE-ZDRW-04, D.W.).

## Author contributions

D.W. designed the research, performed the machine learning, interpreted the results, and led the draft of the manuscript; J.T. carried out data preparation; Z.Z.W., H.Z., W.Z. and X.D.W. contributed to the discussion of the results.

## Competing interests

The authors declare no competing interests.
