## [Peer Review File · Nature Communications]

Elevation-dependent pattern of net CO₂ uptake across ChinaREVIEWER COMMENTS

Reviewer #1 (Remarks to the Author):

Using collected eddy covariance measurements, Wei et al. investigated the elevation-dependent of CO₂ uptake. Then they attributed the differences between the uptake of mountains and that of lower elevation to the temperature sensitivity and human activities. Furthermore, they trained machine learning algorithms to predict the future changes of CO₂ uptakes and found the mountains took higher CO₂ uptakes. This manuscript did some efforts in advancing our understanding on CO₂ uptakes, which was the key issue of climate change, ecology, earth sciences. The topic and findings were interesting and useful for carbon managements at mountains, while the methods were not well documented to persuade the readers believe in these results. Therefore, I believed that this manuscript could not be published before it suffered substantial revisions.

1. The authors emphasized the elevation-dependent of CO₂ uptake across China mountains, while I was not confident with this description. The elevation-dependent did exist but not at China mountains. The elevation-dependent pattern was drawn by collecting 204 eddy covariance measurements. However, the collected eddy covariance measurements were mostly distributed in lower elevation regions. Only 88 sites had an elevation higher than 1000 meters, which was deemed as mountains. From Figure 1b, the elevation-dependent pattern of CO₂ uptake was primarily sourced from their differences between lower elevation regions and mountains. With all data from mountains, whether elevation-dependent pattern existed was questionable. In addition, the elevation-dependent pattern of CO₂ uptake was mainly attributed to temperature sensitivity and human activities, while whether there were other factors like CO₂ concentration?
2. The authors employed temperature sensitivity to address the differences in CO₂ uptake between mountains and lower elevation regions. However, the description about calculating this key parameter was not well documented, which made this explanation hard to justify. I agreed that higher elevations did have higher temperature sensitivity, while the authors should provide detailed methods describing the calculations and the values.
3. The authors used machine learning algorithms to predict the changes in CO₂ uptake under future climate change. However, the data used for training the algorithm were recent data. Extrapolating the current measurements to the future had large uncertainties. In addition, the collected data were mainly the yearly data, while the data used for training random forest algorithms were daily data? What was the source of those data?
4. The terms were mixed used throughout the whole manuscript. What was the difference among net CO₂ uptake, CO₂ uptake, NEP?
5. It was strange to see that R² of GPP and ER were lower than 0.3, which were obviously lower than many previous works.
6. The representativeness of sites may be removed to supporting information, which would make the whole manuscript more concise.
7. Some empirical references should be cited.
8. The dataset had some duplication. For example, Haibei1# and Haibei2# were the same site.

Reviewer #2 (Remarks to the Author):

The study compiles and analyses a large set of data from 204 eddy covariance sites ranging from 1000 up to 5000 masl across Chinese ecosystems and demonstrates a strong elevation dependent CO₂ uptake pattern. Although we know of this pattern for forests shown previously in another gradient, with a different methodology, with less experimental sites, for GPP and NPP but not for NEP in the Peruvian Andes from 0-3000 masl, Fig 3 in Malhi et al 2017, *New Phytologist*, <https://nph.onlinelibrary.wiley.com/doi/epdf/10.1111/nph.14189>), this pattern is demonstrated in the current study with an enormous data set across ecosystems. The implications of the work are of relevance for global carbon cycling, global biogeochemical cycling and global environmental change.

The authors need to elaborate further on what is the temperature gradient across elevations in the study sites. Importantly there is a need to explain the lack of temperature sensitivity in GPP below 3000 m, contrary to what is shown for the Andes in the reference above. The authors have not made any attempt to explain this result.

Few figure legends lack some details that would help to understand the figures. See comments below

In the methods it is unclear if the authors tested for the significance of difference in slopes on the NEP figure and Q10 values obtained from RE on correspondent to Figure 3. The method should explain how the changes in Figs 4c and 5 were calculated.

Not 100% clear how to obtain the flux data in order to reproduce the work.

Fig 1 a

Include what the shades of green and blue in the background mean.

Fig S1a, Include what circle size indicate

Table S1, are these the names of the eddy covariance sites? Should add the word sites or locations to the legend

What does the (x2) (x3) means, two, three towers?

Fig 2a

R² in boxes correspond to relationships between which variables? There are more than one arrow in most cases arriving to a single box. Unclear what the R² corresponds to.

Dashed lines are not easy to differentiate from solid lines

L130 'indicating the dominant role of plant growth in affecting the net CO₂ sink 4' GPP is not plant growth, it is photosynthesis, best to use the correct terminology.

Fig 3, unclear what is the difference is between solid and continuous lines

L151-152

The lack of temperature sensitivity of GPP for ecosystems below 3000 (cite the temperature variation) is an unexpected result, what can explain this?

L155-157

Fig 3 NEP, are the slopes significantly different ?

L148-149 - Are the Q10 significantly different ?

What is the temperature in this elevation gradient?

L162, needs to briefly elaborate on what have been the changes in reactive N in mountains in the study region.

L163 write explicitly the GPP proxy that was used

L173..NEP, RE, or GPP is not plant growth, eddy covariance does not measure plant growth.

L190-191, please indicate where this is shown

Unclear how to access the data, a link is provided, but when typing some sites names or publications from supplementary table nothing came up, authors should clearly explain how to access the data.

Reviewer #3 (Remarks to the Author):

Review comments

The main target of this manuscript is to investigate the elevation dependent CO₂ uptake pattern across China mountains, considering the effects of geographical factors, climate factors, and soil properties. This study reported a negative linear elevation-dependent pattern of CO₂ uptake, and this pattern has been verified by land surface model simulations and satellite observations. This work contributes significantly to a comprehensive understanding of the responses of mountainous ecosystem functioning to the ongoing and future climate change. Despite that the topic is interesting, I have some major concerns which should be clarified before publication.

1. Detailed and accurate climate data are important for the analyses. In this study, the authors used the climate data from the CMFD dataset, which is with a coarse spatial resolution and how does this dataset consider the topography effects on climate data? Will this introduce uncertainty into your analysis?

2. Another issue is regarding the comparison between eddy covariance data and MsTMIP simulations. The spatial resolution is 0.5 degree for MsTMIP models, which does not match the detailed observations

of EddyChina2023. Again, the CMIP6 model projections are with even much coarser resolution for the future climate change, so what is the uncertainty for comparing the future climate change on high elevation and low elevation?

3. One main finding of this study is that ecosystems on high mountains have higher temperature sensitivity and future climate change will increase the CO₂ uptake in these ecosystems. However, I found weak relationship between either GPP or RE and air temperature from the main figure 2, which is the most important figure illustrating the relationship between different components of NEP and multiple factors. There seems to be contradictory between them.

Detailed comments

1. line 37-42, This part is not relevant to the topic of this manuscript.
2. Line 98-103, In which period are these results determined? Do you consider the potential difference in the interannual variations in NEE among different sites in different ecosystems?
3. Another point is that the occurrence of extreme climate events may be quite different
4. Line 106, In this study, the largest uptake of CO₂ is found in croplands, but not forests.
5. Line 112, This sentence is not clear. The CO₂ uptake in temperate grasslands of Inner Mongolia is neutral, indicating that the carbon gain is cancelled out by the carbon loss. How can this result be verified by soil carbon inventory? Do you mean repeated inventory data?
6. Figure 2, It is strange that the air temperature does not exert important effects on either GPP or RE across mountainous sites.
7. Line 135-137, But from the SEM analysis, I did not find a close linkage between NEP and variations of temperature and precipitation and nitrogen deposition.
8. Figure S14, What do the different colored lines represent?

Response Letter

Reference Number: NCOMMS-23-30115T

Title: Elevation-dependent pattern of net CO₂ uptake across China

To Reviewer #1

Using collected eddy covariance measurements, Wei et al. investigated the elevation-dependent of CO₂ uptake. Then they attributed the differences between the uptake of mountains and that of lower elevation to the temperature sensitivity and human activities. Furthermore, they trained machine learning algorithms to predict the future changes of CO₂ uptakes and found the mountains took higher CO₂ uptakes. This manuscript did some efforts in advancing our understanding on CO₂ uptakes, which was the key issue of climate change, ecology, earth sciences. The topic and findings were interesting and useful for carbon managements at mountains, while the methods were not well documented to persuade the readers believe in these results. Therefore, I believed that this manuscript could not be published before it suffered substantial revisions.

Dear Reviewer,

Thanks very much for your encouragement on the novelty of our work. We also appreciate your time and efforts to improve this manuscript. During the revision, all your concerns have been carefully considered, explained, and resolved.

- **Elevations.** According to your suggestion, we covered all elevation bands rather than mountains only in the revised manuscript. Several figures have been redrawn and many words have been revised, including the Title, i.e., the title has been changed to “Elevation-dependent pattern of net CO₂ uptake across China”, by deleting the “mountainous China”.
- **SEM analysis.** To improve the performance of SEM analysis, we updated the SEM, including using ground-based climate data, adding more dependent variables, and re-organizing the model structure. These did effectively improve the performance of SEM analysis. Also, we provided more details regarding the calculation and statistics of temperature sensitivity. For Fig. 3, a new section named “Temperature sensitivity” was also added to “Method and materials”.
- **More concise.** Following your suggestion, to make the whole manuscript more concise, we moved the “*Representativeness of the eddy covariance dataset*” to the Supporting Materials (Text S1).
- **Terms.** Regarding the terms, we made them consistency throughout the manuscript, i.e., change “CO₂ uptake”, “CO₂ sink”, “net CO₂ uptake” to “NEP”.
- **Dataset.** Your concerns regarding dataset have also been resolved, by uploading the dataset to the website Figshare (<https://figshare.com/s/e1f7f9c13e547e422a71>).

Thanks very much for your efforts to improve our manuscript. We learned a lot from your work.

Best regards,

The authors

1. The authors emphasized the elevation-dependent of CO₂ uptake across China mountains, while I was not confident with this description. The elevation-dependent did exist but not at China mountains. The elevation-dependent pattern was drawn by collecting 204 eddy covariance measurements. However, the collected eddy covariance measurements were mostly distributed in lower elevation regions. Only 88 sites had an elevation higher than 1000 meters, which was deemed as mountains. From Figure 1b, the elevation-dependent pattern of CO₂ uptake was primarily sourced from their differences between lower elevation regions and mountains. With all data from mountains, whether elevation-dependent pattern existed was questionable. In addition, the elevation-dependent pattern of CO₂ uptake was mainly attributed to temperature sensitivity and human activities, while whether there were other factors like CO₂ concentration?

Response:

Thanks for your comments here.

- We agree with you that **the elevation-dependent did exist but not at China mountains**. Indeed, the eddy covariance, MsTMIP models and remote sensing observations consistently support the elevation-dependent pattern of CO₂ exchanges. Therefore, several revisions were made to reflect the elevation dependent pattern of net CO₂ uptake in China, rather than China's mountains. (1) We have revised the title to "*Elevation-dependent pattern of net CO₂ uptake across China*". The "mountainous China" has been revised to "China". (2) We include the elevation of 0-1000 m in all figures and redraw Figure 3, 4, 5, S8 and S14. The word "Mountains" has been deleted from all figures. (3) In the main text, we shorten the descriptions regarding mountains to cover all elevation bands across China, rather than mountains only.
- Many environmental drivers affect the elevation-dependent pattern of NEP, including climate factors, nutrient supplies, human activities, and atmospheric CO₂ concentration (denoted by [CO₂]). In the populated lower elevation region, the higher CO₂ concentration may benefit photosynthesis via the CO₂ Fertilization Effect (CFE). Therefore, we added the [CO₂] into the SEM analyses, though it turned out that its contribution to the variation on GPP did not reach a significant level (**Fig. 2a**). This is also consistent to the correlation analyses, i.e., the correlation between GPP and [CO₂] does not reach a significant level (**Fig. S5**). This indicates that the contribution of [CO₂] to the spatial pattern of NEP may be relatively lower, compared with the climatic factors, ecosystem types and nitrogen deposition.
- We thank you for your efforts to improve our manuscript.

2. The authors employed temperature sensitivity to address the differences in CO₂ uptake between mountains and lower elevation regions. However, the description about calculating this key parameter

was not well documented, which made this explanation hard to justify. I agreed that higher elevations did have higher temperature sensitivity, while the authors should provide detailed methods describing the calculations and the values.

Response:

- Thanks for your comment regarding the calculation of temperature sensitivity. Following your suggestion, we added a new section in the Material and Method, namely “*Temperature sensitivity*”, which provide detailed description regarding the definition and calculation. We also made additional statistical analyses, showing significant difference in temperature sensitivities between different elevation bands. The above statistical results were then clarified in the main text (**Line 122, Line 124, and Line 127**).
- To facilitate your review, we also put the descriptions here. “*For GPP and NEP, their temperature sensitivities were defined as the slopes of their variations on air temperature with linear regressions. For RE, the Van't Hoff equation ($y = ae^{bt}$) was used to calculate the temperature sensitivity ($Q_{10} = e^{10b}$). During the calculation, the daily average aboveground air temperature at 2 m was obtained from the CMFD dataset, given most towers did not provide air temperature observations in the literature. We then used General Linear Model (GLM) univariate analysis to test the differences regarding the regression lines for GPP, RE and NEP. During this process, the GPP, RE and NEP were the dependent variables, the different elevation groups (higher or lower than 3000 m in elevation) were the independent variables, while air temperature was covariate. For GPP, over 40,000 observations were involved in the GLM analysis, while there were more than 50,000 observations for RE and more than 60,000 observations for NEP. Daily accumulated or averaged GPP, NEP and air temperature data were used during the GLM analyses. For RE, given exponential regression models were used to reflect its temperature sensitivities, we applied a log transformation to the RE data before the GLM analysis. Effects were considered as significant at $P < 0.05$ (difference in temperature sensitivities between elevation bands), while the null hypothesis was accepted when $P > 0.05$ (no difference in temperature sensitivities between elevation bands)*”. Please see **Line 285 to 300**.
- We hope the descriptions regarding the definition, calculation and statistical analyses would make the manuscript more readable. Thanks very much for helping us with improving the manuscript.

3. The authors used machine learning algorithms to predict the changes in CO₂ uptake under future climate change. However, the data used for training the algorithm were recent data. Extrapolating the current measurements to the future had large uncertainties. In addition, the collected data were mainly the yearly data, while the data used for training random forest algorithms were daily data? What was the source of those data?

Response:

- Yes, we used the **daily data** to train and validate the random forest algorithms. The algorithms retrieved from the random forest training was then used to predict future variation in NEP under

various climate scenarios. We apology that the original link to the daily data of EddyChina2023 did not work. To resolve this issue, we then uploaded the dataset to the website Figureshare (<https://figshare.com/s/e1f7f9c13e547e422a71>). We hope the reviewers and potential readers would get access to the dataset easily.

- Before predicting future variation in NEP with random forest, the results of random forest training based upon EddyChina2023 dataset is compared with the FLUXCOM regarding its seasonal pattern and magnitude (**Fig. R1**) (Jung, Schwalm et al. 2020). The FLUXCOM is a global scale upscaling product based upon FLUXNET2015. The NEP outputs of random forest training also in line with previous upscaling and remote sensing studies in China (Yao, Li et al. 2018, Chen, Park et al. 2019).
- Still, we agree that using the current dataset to predict future variation in NEP would cause uncertainty. That is why we combined the eddy covariance dataset and the CMIP6 outputs with the help of random forest, to constrain future variation in the NEP. Also, the CMIP6 outputs in the current study include 4 climate scenarios and each scenarios include over 20 models. We hope the current method, i.e., combine of EddyChina2023 and CMIP6, would better constrain the future variation in NEP under future climate changes.
- In addition, all climate scenarios suggest that higher elevation environments would experience more robust changes in NEP, compared with the lower elevation environments. The random forest predications are generally consistent to the results based upon the GLASS LAI product, both highlighting the importance of human impacts on decadal variation in net CO₂ uptakes at lower elevation bands, while high elevation environments would be more productive under climate changes.

References

- Chen, C. *et al.* China and India lead in greening of the world through land-use management. *Nat. Sustain.* **2**, 122-129 (2019). <https://doi.org/10.1038/s41893-019-0220-7>
- Jung, M. *et al.* Scaling carbon fluxes from eddy covariance sites to globe: synthesis and evaluation of the FLUXCOM approach. *Biogeosciences* **17**, 1343-1365 (2020). <https://doi.org/10.5194/bg-17-1343-2020>
- Yao, Y. T. *et al.* A new estimation of China's net ecosystem productivity based on eddy covariance measurements and a model tree ensemble approach. *Agricultural and Forest Meteorology* **253**, 84-93 (2018). <https://doi.org/10.1016/j.agrformet.2018.02.007>

Fig. R1 Spatial pattern, trend, and magnitude of NEP across China's terrestrial ecosystems based upon the random forest training of EddyChina2023 dataset. (a) Averaged NEP of China's terrestrial ecosystems during the past two decades. **(b)** Trend of annual NEP of China's terrestrial ecosystems. **(c)** Comparison between monthly NEP of China's terrestrial ecosystems, estimated by EddyChina2023 and FLUXCOM. The EddyChina2023-RF represents the results of random forest training based upon the EddyChina2023 dataset.

4. The terms were mixed used throughout the whole manuscript. What was the difference among net CO₂ uptake, CO₂ uptake, NEP?

Response:

Thanks very much for your comment regarding the terms. We apologize that our terms may have caused misunderstanding in a previous version of the manuscript. Following your suggestion, we

carefully revised all terms regarding the CO₂ uptakes throughout the whole manuscript, including the title, main text, figure captions and supporting materials. Here we list how these terms were corrected in the revised manuscript.

- We clarify at the beginning of the Introduction that “*net uptake of CO₂ is denoted by net ecosystem productivity, NEP*”. In the main text, most “net CO₂ uptake” have been replaced with NEP, except the title.
- If necessary, we keep terms “uptake of atmosphere CO₂” and “net CO₂ sink”. For example, “Existing studies have highlighted that these mountains function as a vital *net CO₂ sink*”.
- Thanks again for your comment regarding the terms, which make the manuscript more formal and concise.

5. It was strange to see that R² of GPP and RE were lower than 0.3, which were obviously lower than many previous works.

Response:

Thanks very much for concern regarding lower R² for either GPP or RE.

- For a specific eddy covariance site, the **temporal pattern** of GPP and RE are usually well explained by abiotic and biotic factors like temperature, radiation, precipitation, soil moisture and leaf area index, which have been seen by lots of studies. For example, in one alpine wetland site run by ourself, the temporal variations in GPP and RE were well explained by climate factors, with R²=0.70 and 0.75, respectively (Qi et al., 2021, JGR).
- We are also wondering whether we have missed some important processes that govern the **spatial pattern** of GPP and RE (**Figure R2a**). We questioned that the situation you mentioned would be improved if more variables, like ecosystem types and atmospheric CO₂ concentration, were added to the SEM. In addition, we also employed the ground-based climate data for each site (from the literature of each eddy covariance tower), rather than the CMFD dataset. It seems that adding more variables, using the ground-based climate data, and re-organizing the model structure did improve the performance of SEM analysis (**Figure R2a**). **The R² for GPP has been enhanced to 0.39, while the R² for RE has been enhanced to 0.29.**
- Still the R² for GPP and RE are much lower than our expectation. The situation is quite similar for the spatial pattern of GPP, RE and NEP along the elevation (R range from -0.18 to -0.32). We consider the relatively lower R² or R for GPP and RE may due to the large spatial range of the EddyChina2023 dataset., i.e., it covers a latitude-span of 40 degree (15°N to 55°N). Therefore, the spatial patten of GPP and RE (elevation-dependent pattern in this study) may include unrecognized contributions from other unrecognized sources, like forest species, management (e.g., irrigation), observations (e.g., different equipment and setting) and calculation (e.g., data processing and quality control), though their impacts can hardly be determined in the SEM.

Fig. R2. Controls on the elevation-dependent pattern of NEP. (a) Indirect and direct impacts of geographical factors, climate factors and soil processes on ecosystem CO₂ fluxes. GPP, gross primary productivity; RE, ecosystem respiration; NEP, net ecosystem productivity; PAR, photosynthetically active radiation; TAS, atmospheric temperature; Nr, reactive nitrogen; PREC, precipitation; CO₂, atmospheric CO₂ concentration. Given NEP represents the net balance between GPP and RE, we built its indirect relationship with environmental factors via GPP and RE. The structural equation model analysis was calculated based upon the annual average CO₂ fluxes (multi-year average used if there was more than one year of observations). The blue lines indicate negative impacts, while the red lines indicate positive impacts. The numbers adjacent to the arrows are the standardized path coefficients. The solid lines represent a significant correlation (* $P < 0.05$, ** $P < 0.01$, *** $P < 0.001$), whereas the dashed lines indicate lack of significance. The R^2 values in the boxes indicate how much the dependent variables are explained by the independent variable(s). (b) Variation in TAS, PREC and Nr with elevation. The dashed lines indicate the linear or nonlinear fit for the environmental factors, whereas the gray shading represents the 95% confidence band of the fits.

Reference: Yahui Qi, Da Wei*, Hui Zhao, Xiaodan Wang* (2021) Carbon sink of a very high marshland on the Tibetan Plateau. *Journal of Geophysical Research: Biogeosciences*. 126, e2020JG006235.

6. The representativeness of sites may be removed to supporting information, which would make the whole manuscript more concise.

Response: Thanks for your suggestion. We have now moved this section to Supporting Materials and named it Text S1. This made the whole manuscript more concise and easier to follow for readers.

7. Some empirical references should be cited.

Response:

Thanks for your suggestion regarding empirical references. We read more papers on elevation dependent and carbon flux. We found lots of studies have been conducted in Alps, Andes, and the Tibetan Plateau, especially in the Alps. Therefore, these references were added to the main text: "Several studies have been conducted across Earth's high mountains, i.e., Alps(Zeeman, Hiller et al. 2010), Andes(Malhi, Girardin et al. 2017), Tibetan Plateau(Zhao, Li et al. 2017) and etc., suggesting varying elevation-dependent patterns of GPP and RE."

References:

Zeeman, M. J. et al. Management and climate impacts on net CO₂ fluxes and carbon budgets of three grasslands along an elevational gradient in Switzerland. *Agricultural and Forest Meteorology* 150, 519-530 (2010).

Malhi, Y. et al. The variation of productivity and its allocation along a tropical elevation gradient: a whole carbon budget perspective. *New Phytologist* 214, 1019-1032 (2017).

Zhao, J., Li, R., Li, X. & Tian, L. Environmental controls on soil respiration in alpine meadow along a large altitudinal gradient on the central Tibetan Plateau. *Catena* 159, 84-92 (2017).

8. The dataset had some duplication. For example, Haibei1# and Haibei2# were the same site.

Response:

- Indeed, the data collection from literatures is not an easy job. For a specific station, it may include several eddy covariance towers, to cover several ecosystem types. For example, in Haibei Station, at least three ecosystem types were covered by eddy covariance towers, i.e., alpine meadow, alpine shrub and wetland.
- When there are two or more publications regarding the eddy covariance observation for each ecosystem type and for each station, we judge whether they are the same one by checking their author list and whether they have the same instruments.
- We reviewed roughly 400 papers to obtain the EddyChina2023 dataset, and we apologize that we may have made a mistake regarding the Haibei Station. We thank the reviewer's comment that the Haibei#1 and Haibei#2 may be the same site. During the revision, we rectified the above mistake. In addition, the number "204" has been corrected to "203" throughout the manuscript, including Main text, Figures, Supporting Materials and Datasets.

To Reviewer #2

The study compiles and analyses a large set of data from 204 eddy covariance sites ranging from 1000 up to 5000 masl across Chinese ecosystems and demonstrates a strong elevation dependent CO₂ uptake pattern. Although we know of this pattern for forests shown previously in another gradient, with a different methodology, with less experimental sites, for GPP and NPP but not for NEP in the Peruvian Andes from 0-3000 masl, Fig 3 in Malhi et al 2017, *New Phytologist*, <https://nph.onlinelibrary.wiley.com/doi/epdf/10.1111/nph.14189>), this pattern is demonstrated in the

current study with an enormous data set across ecosystems. The implications of the work are of relevance for global carbon cycling, global biogeochemical cycling, and global environmental change.

Dear Reviewer,

Thank you very much for reviewing the paper and for providing an excellent literature. All your concerns have been carefully considered and revised, and here is brief response to your comments.

- **National scale study.** Our research involves over 200 eddy covariance sites across China, from the tropics to the cold zone (temperature span exceeds 30°C degrees), and some ecosystems involve lots of human activities (China's population density below 3,000 meters is very high). The complexity of China means that our results may be slightly different from the study in Peru. We honestly present these data in the manuscript, albeit somewhat difference from the study in Peru. We think the comparison between these works can deepen our understanding of the elevation dependent pattern of CO₂ uptakes.
- **Terms.** You raised questions about terminology in your paper, such as GPP not plant growth. We thank you for pointing out these errors, and we have corrected them based on your comments.
- **Figure captions.** You mentioned that there are too few descriptions in some figures, making it very difficult for reviewers or potential readers to understand the results. At the same time, you raised some questions about statistics in your paper. We are very grateful to you for pointing out these important issues, and we have added them in the revised paper, especially Figure 3, Figure 4, Figure 5 and the supporting figures.
- **Dataset.** We are sorry that the link we submitted previously seems to have expired. We have re-uploaded the data to Figshare (<https://figshare.com/s/e1f7f9c13e547e422a71>), and we hope that all data can be reused easily.
- **Minor comments.** Your minor, details comments have been carefully considered and revised in the revised manuscript.

Thanks again for your efforts to improve this manuscript.

Best regards,

The authors

The authors need to elaborate further on what is the temperature gradient across elevations in the study sites. Importantly there is a need to explain the lack of temperature sensitivity in GPP below 3000 m, contrary to what is shown for the Andes in the reference above. The authors have not made any attempt to explain this result.

Response:

- Thanks for your comment regarding the temperature sensitivity.
- In fact, temperature strongly affect **spatial pattern** of GPP (**Fig. S5**), which is consistent to the

result in Peru (i.e., temperature largely affect the spatial pattern of GPP). Regarding the temperature sensitivity of GPP for below 3000 m, temperature still plays a role in regulating its **temporal variation**, i.e., $P < 0.001$, given more than 30000 observations were involved in the calculation, though the R^2 is only 0.04 (or $R = 0.20$).

- For the reasons, we consider the higher temperature and strong human activities in the lower elevation environments may have contributed to lower temperature sensitivity of GPP, than that in high elevation environments. **(1)** The mean annual temperature for the ecosystems below 3000 m is 10.9°C (-4.4 to 25.0°C), while it is -0.12°C (-6.0 to 8.7°C) for the ecosystems above 3000 m. The high elevation environments (3000-5000 m) are 11°C colder than the ecosystems below 3000 m, which may have caused the large difference in temperature sensitivity in GPP, RE and NEP. **(2)** Our study covers a large geographic extent across China (15°N - 55°N), including rubber plantation in Hainan Island, rice paddies in Yangtze River Delta, maize & corn in North China Plain, and soybean in Northeast China. As a highly populated country, these regions experienced strong human management like irrigation and fertilization. Management of water and nutrient in the highly-populated region (< 3000 m) may have caused a relatively weaker role of temperature on regulating the temporal variation in GPP. Also, the RE and NEP also show lower temperature sensitivity, which is consistent to the GPP.
- Trying to resolve your concerns, we now have added the following sentences in the main text: “Further analyses validate our expectation that high-elevation environments and thus colder ecosystems (3000–5000 m; mean annual temperature of -0.12°C , ranging from -6.0 to 8.7°C), such as the Tibetan Plateau, are more sensitive to climate warming than their lower-elevation counterparts (mean annual temperature of 10.9°C , ranging from -4.4 to 25.0°C)”. Please see **Line 117-120**. “*The high elevation environments is 11°C colder than the ecosystems below 3000 m, which may have caused the large difference in temperature sensitivity in GPP. Beside the variation in temperature, management of water and nutrient in highly-populated region (< 3000 m) may also strongly regulate the variation in plant photosynthesis*”. Please see **Line 124 to 128**.
- Thanks for your comments regarding the temperature sensitivity.

Few figure legends lack some details that would help to understand the figures. See comments below.

Response: Thanks for your comments regarding figure legends. To resolve your concerns, we then added more descriptions in each figure. More descriptions are also presented in Method and Materials. We hope the additional information would make the whole manuscript more friendly to reviewer and potential readers.

In the methods it is unclear if the authors tested for the significance of difference in slopes on the NEP figure and Q10 values obtained from RE on correspondent to Figure 3. The method should explain how the changes in Figs 4c and 5 were calculated.

Response: Thanks for your comment regarding the Figure 3-5. During the revision, we added more description to resolve your concern, which are listed here.

- For **Figure 3**, a new section named “**Temperature sensitivity**” was added to the Method and Materials: “For GPP and NEP, their temperature sensitivities were defined as the slopes of their variations on air temperature with linear regressions. For RE, the Van't Hoff equation ($y = ae^{bt}$) was used to calculate the temperature sensitivity ($Q_{10} = e^{10b}$). During the calculation, the daily average aboveground air temperature at 2 m was obtained from the CMFD dataset, given most towers did not provide air temperature observations in the literature. We then used General Linear Model (GLM) univariate analysis to test the differences regarding the regression lines for GPP, RE and NEP. During this process, the GPP, RE and NEP were the dependent variables, the different elevation groups (higher or lower than 3000 m in elevation) were the independent variables, while air temperature was covariate. For GPP, over 40,000 observations were involved in the GLM analysis, while there were more than 50,000 observations for RE and more than 60,000 observations for NEP. Daily accumulated or averaged GPP, NEP and air temperature data were used during the GLM analyses. For RE, given exponential regression models were used to reflect its temperature sensitivities, we applied a log transformation to the RE data before the GLM analysis. Effects were considered as significant at $P < 0.05$ (difference in temperature sensitivities between elevation bands), while the null hypothesis was accepted when $P > 0.05$ (no difference in temperature sensitivities between elevation bands)”. Please see **Line 285 to 300**.
- In **Figure 4**, we added more descriptions in figure legends to explain the calculation of the mean GPP, LAI and their relative changes (Δ GPP and Δ LAI): “**Fig. 4. Elevation-dependent pattern of gross primary productivity (GPP), leaf area index (LAI) and their changes during the last two decades.** (a) Elevation-dependent pattern of the annual cumulative GPP with the MsTMIP model ensemble. For the MsTMIP ensemble, all pixels of each elevation band were employed to derive the annual cumulative GPP for each model. Then, the results of these models were averaged to obtain the ensemble average GPP and the standard deviation for each elevation band. (b) Elevation-dependent pattern of LAI from the GLASS LAI product³⁶. (c) Relative changes in the GPP and LAI along the elevation gradient. For the V_{GPP} , it was used to represent the speed of variation in GPP along the elevation gradient. The V_{GPP} was calculated by dividing the slope of GPP changes by the average GPP: $V_{GPP} = \text{Slope}(\text{GPP}) / \text{GPP}_{\text{baseline}}$. In this equation, $\text{GPP}_{\text{baseline}}$ represents the annual average GPP during 2000–2004 (the first five years). The line extending from each bar represents the uncertainty in each climate scenario (± 1 SD) of the MsTMIP model group (eight models: BIOME-BGC, CLASS-CTEM-N, CLM4, CLM4VIC, DLEM, ISAM, TEM6 and TRIPLEX-GHG). For V_{LAI} , a similar procedure was employed by calculating the relative changes, i.e., the slope of LAI change divided by the average LAI in baseline years: $V_{LAI} = \text{Slope}(\text{LAI}) / \text{LAI}_{\text{baseline}}$, where $\text{LAI}_{\text{baseline}}$ represents the annual average GPP during 2000–2004 (the first five years)” Please see **Line 571 to 585**.
- In **Figure 5**, similar calculation procedure was used to calculate the Δ NEP and more information

was added to the figure legends: “**Fig. 5. Relative changes in net ecosystem productivity (Δ NEP) under various climate scenarios constrained by the CMIP6 models and the EddyChina2023 dataset.** Relative changes in NEP under (a) SSP1-2.6, (b) SSP2-4.5, (c) SSP3-7.0 and (d) SSP5-8.5. Relative changes were calculated by dividing the absolute changes in NEP by their average for 2020–2024 (the first five years): Δ NEP_i=(NEP_i-NEP_{baseline})/NEP_{baseline}. Here, NEP_i represents the NEP in year *i*, while NEP_{baseline} represents the averaged NEP during 2020–2024. During the calculation, all pixels of each elevation band were employed to derive the annual cumulative GPP for each model under each climate scenario. Then, the results of these models were averaged to obtain the ensemble average Δ NEP and the standard deviation for each elevation band under each climate scenario. Solid curves are the ensemble mean of the model simulations and the shading represents ± 1 SD. The bar charts represent the relative changes by the end of the research period (2056–2065), within which the line extending from each bar represents the uncertainty in each climate scenario (± 1 SD)”. **Please see Line 587 to 598.**

- Thanks for your comments regarding the above figures.

Not 100% clear how to obtain the flux data in order to reproduce the work.

Response:

We apologize that the original link to the EddyChina2023 dataset does not work. To resolve this issue, we uploaded the dataset to the website of *figshare* (<https://figshare.com/s/e1f7f9c13e547e422a71>). The new address of the dataset is also added to *Data Availability*.

Fig 1 a

Include what the shades of green and blue in the background mean.

Response:

- In a previous version of the Figure 1a, we used the “Green and Blue” colors in the 3D map, to illustrate the mountainous China and location of the eddy covariance towers. The shades of green and blue is used after traditional Chinese painting.
- Your another comment regarding Table S1 suggest adding site name into the map, we then re-drawn the map of **Figure 1a** (in the main text) with a 2D map, which is also shown here (**Fig. R3**).

Figure R3. Locations of eddy covariance observation sites across China. To simplify the map, we only list the main ecosystem type for each station, though several towers may have been established to cover multiple ecosystem types for some stations (for details, see Table S1).

Fig S1a, include what circle size indicate

Response: Thanks for your comments. Follow your suggestion, we have added a new legend to Figure S1a. The captions of the **Figure S1a** has also been revised to indicate “the size of the circle represents the observation years for each site”.

Table S1, are these the names of the eddy covariance sites? Should add the word sites or locations to the legend. What does the (x2) (x3) means, two, three towers?

Response:

- Yes, the Table S1 provide the exact site names and the tower number for each station. The x2 and x3 indicates two and three towers for each station, respectively, given there may be several ecosystem types for a station.
- After your suggestion, we added the sites and locations into the Figure 1a. Please see **Line 205** in Supporting Materials.
- Thanks for your suggestion to help us provide a much better illustration of the flux towers.

Fig 2a. R2 in boxes correspond to relationships between which variables? There are more than one arrow in most cases arriving to a single box. Unclear what the R2 corresponds to. Dashed lines are

not easy to differentiate from solid lines

Response:

- The R^2 in each box represents how much the dependent variable is explained by independent variables. For example, the spatial variation of TAS (air temperature) is 97% explained by latitude and elevation.
- We redraw the Figure 2a to make the dash lines different from the solid lines. Other details regarding the Figure 2a have also been revised to make the whole figure more readable.
- Thanks very much for your comments regarding this figure.

L130 'indicating the dominant role of plant growth in affecting the net CO₂ sink 4' GPP is not plant growth, it is photosynthesis, best to use the correct terminology.

Response: Thanks for your comment. Now the sentence has been revised to "..... *indicating the dominant role of photosynthesis in affecting the NEP*". Please see **Line 104**.

Fig 3, unclear what is the difference is between solid and continuous lines

Response:

- We redraw the **Figure 2a** to make the dash lines different from the solid lines.
- The solid lines represent a significant correlation (* $P < 0.05$, ** $P < 0.01$, *** $P < 0.001$), whereas the dashed lines indicate lack of significance.

L151-152

The lack of temperature sensitivity of GPP for ecosystems below 3000 (cite the temperature variation) is an unexpected result, what can explain this?

Response: Thanks for your comment here.

- Regarding the temperature sensitivity of GPP for below 3000 m, temperature still plays a role in regulating its temporal variation, i.e., $P < 0.001$, given more than 30000 observations were involved in the calculation, despite that the R^2 is only 0.04 (or $R = 0.20$). This indicates the temperature may play a minor role in explaining the variation in GPP in these ecosystems.
- We consider the higher temperature and strong human activities in the lower elevation environments may have contributed to lower temperature sensitivity of GPP, than that in high elevation environments. **(1)** The mean annual temperature for the ecosystems below 3000 m is 10.9°C (-4.4 to 25.0°C), while it is -0.12°C (-6.0 to 8.7°C) for the ecosystems above 3000 m. The high elevation region (>3000 m) is 11°C colder than the ecosystems below 3000 m, which may have caused the large difference in temperature sensitivity in GPP, RE and NEP. **(2)** It is notable that the ecosystems below 3000 m cover a large geographic extent across China, from tropical forests in Hainan Island, rice paddies in Yangtze River Delta, maize & corn in North China Plain, and soybean in Northeast China. As a highly populated country, these regions experienced strong human impacts like irrigation and fertilization. Management of water and

nutrient in highly-populated region (<3000 m) may also strongly regulate the variation in plant photosynthesis. Also, the RE and NEP also show lower temperature sensitivity, which is consistent to the GPP.

- We now have added the following sentences in the main text. "Further analyses validate our expectation that high-elevation environments and thus colder ecosystems (3000–5000 m; mean annual temperature of -0.12°C , ranging from -6.0 to 8.7°C), such as the Tibetan Plateau, are more sensitive to climate warming than their lower-elevation counterparts (mean annual temperature of 10.9°C , ranging from -4.4 to 25.0°C ". Please see **Line 117-120**. "*The high elevation environments is 11°C colder than the ecosystems below 3000 m, which may have caused the large difference in temperature sensitivity in GPP. Beside the variation in temperature, management of water and nutrient in highly-populated region (<3000 m) may also strongly regulate the variation in plant photosynthesis*". Please see **Line 124 to 128**.
- Thanks for your comments regarding the temperature sensitivity of GPP.

L155-157

Fig 3 NEP, are the slopes significantly different?

Response:

- Thanks for your comment. Yes, our additional analyses suggest that there is significant difference between the higher and lower elevation regions regarding their temperature sensitivities ($P < 0.01$), consistently for GPP, RE and NEP. We have added the significant level to the main text. Please see **Line 122, 124 and 127**.
- We also provided descriptions of the calculation and statistics in the "Temperature sensitivity" of the Material and methods: "*We then used General Linear Model (GLM) univariate analysis to test the differences regarding the regression lines for GPP, RE and NEP. During this process, the GPP, RE and NEP were the dependent variables, the different elevation groups (higher or lower than 3000 m in elevation) were the independent variables, while air temperature was covariate. For GPP, over 40,000 observations were involved in the GLM analysis, while there were more than 50,000 observations for RE and more than 60,000 observations for NEP. Daily accumulated or averaged GPP, NEP and air temperature data were used during the GLM analyses. For RE, given exponential regression models were used to reflect its temperature sensitivities, we applied a log transformation to the RE data before the GLM analysis. Effects were considered as significant at $P < 0.05$ (difference in temperature sensitivities between elevation bands), while the null hypothesis was accepted when $P > 0.05$ (no difference in temperature sensitivities between elevation bands)*". Please see **Line 290 to 300**.

L148-149 - Are the Q10 significantly different?

What is the temperature in this elevation gradient?

Response:

- Thanks for your comment regarding the statistics of Q_{10} values. Yes, the Q_{10} values between different groups reached a significant level ($P < 0.01$), given more than 50000 observations were employed during the calculation and statistical analyses. We then added the significance level to the main text ($P < 0.01$). Please see **Line 122**.
- Regarding the temperature gradient, we furthered calculated the mean annual temperature of these sites. It turned out that the warmer area (<3000 meters in elevation) has a mean annual temperature at 10.9°C (ranged from -4.4 to 25°C), while the colder area r area (3000 meters in elevation) has the mean annual temperature of -0.12°C (ranged from -6 to 8.7°C). Indeed, there is 11.0C difference between two elevation gradient, which may have caused the significant difference in GPP, RE and NEP regarding their temperature sensitivities. We also added the above information regarding temperature to the main text to facilitate your review: “*Further analyses validate our expectation that high-elevation environments and thus colder ecosystems (3000–5000 m; mean annual temperature of -0.12°C, ranging from -6.0 to 8.7°C), such as the Tibetan Plateau, are more sensitive to climate warming than their lower-elevation counterparts (mean annual temperature of 10.9°C, ranging from -4.4 to 25.0°C).*” Please see **Line 117 to 120**.
- Thanks again for your comment. We hope our revisions have resolved your concern.

L162, needs to briefly elaborate on what have been the changes in reactive N in mountains in the study region.

Response: Thanks for your comment. Following your suggestion, we added another sentence to explain what happened to N deposition in China’s mountains -- “*It is notable that reactive N deposition has increased persistently across high-elevation environments in China²¹. This is different from lower elevations, where N deposition increased (1980–2000) and then stabilized (after 2005), driven by socioeconomic changes and vigorous controls on N pollution^{22,23}.*” Please see **Line 139 to 140**.

L163 write explicitly the GPP proxy that was used

Response: We used the leaf area index as a proxy to GPP. Now the sentence has been revised to “.....*we focused on GPP and its proxy (LAI, Leaf Area Index).*” Thanks for your comment. Please see **Line 141**.

L173..NEP, RE, or GPP is not plant growth, eddy covariance does not measure plant growth.

Response: Thanks for your comment. Now the sentence has been revised to “*similar to the elevation-dependent pattern of GPP based on both the EddyChina2023 dataset and the MsTMIP ensemble*”. Please see **Line 152-153**.

L190-191, please indicate where this is shown

Response: Thanks for your comment. The results are shown in **Fig. 5a-d**. The sentence has now been revised to “*Future changes in climate and nutrient supply will cause a general increase in NEP*”

across most elevation bands and the various climate scenarios (Fig. 5a-d)". Please see Line 167-169.

Unclear how to access the data, a link is provided, but when typing some sites names or publications from supplementary table nothing came up, authors should clearly explain how to access the data.

Response: We apology that the original link to the EddyChina2023 dataset does not work. To resolve this issue, we uploaded the dataset to *FigureShare* (<https://figshare.com/s/e1f7f9c13e547e422a71>). We spent roughly 6 months to create the EddyChina2023 dataset, and hope the dataset could be reused by the scientific community, especially for machine and deep learning. The new address of the dataset has also been added to the section of *Data Availability*. Please see **Line 372**.

To Reviewer #3

The main target of this manuscript is to investigate the elevation dependent CO₂ uptake pattern across China mountains, considering the effects of geographical factors, climate factors, and soil properties. This study reported a negative linear elevation-dependent pattern of CO₂ uptake, and this pattern has been verified by land surface model simulations and satellite observations. This work contributes significantly to a comprehensive understanding of the responses of mountainous ecosystem functioning to the ongoing and future climate change. Despite that the topic is interesting, I have some major concerns which should be clarified before publication.

Dear Reviewer,

Thanks very much for your time and thoughtful comments regarding the manuscript. We are happy to see that our work *contributes significantly to a comprehensive understanding of the responses of mountainous ecosystem functioning to the ongoing and future climate change*. During the revision, we carefully considered all your comments and made lots of revisions, which are summarized here.

- **Climate Dataset.** To resolve your concern regarding climate dataset, we then retrieved ground-based climate data for each site (from the literatures), and compare them with the CMFD dataset (0.1°x0.1°). We also employed a finer climate dataset, namely WorldClim (0.0083°x0.0083°), to explore whether an increase in spatial resolution would reduce its uncertainty. The results generally show that the CMFD is good enough to represent the spatial variation in climate factors.
- **SEM analyses.** Your another concern is about the SEM analyses. In the revision, we updated the SEM by employing the ground-based climate data, rather than the CMFD data. We also added ecosystem types and atmosphere CO₂ concentration to the SEM. The updated SEM analyses show general consistent result to Pearson correlation analyses, i.e., both verified the importance of air temperature in affecting the spatial pattern of GPP. Also, the updated SEM also performs better in explaining the spatial variation in GPP and RE.
- **Uncertainty.** We admit that there is still uncertainty regarding the drivers of the spatial pattern of NEP. Given our study represent a national scale study, which means that other factors may also play a significant role in affecting the *apparent* CO₂ exchange recorded by eddy covariance towers, such as equipment setting, data processing, human activities, species and etc. However,

these unrecognized factors and their contribution can hardly be determined by the present study. Also, the description regarding the uncertainty were also added to the **Implications** of the main text.

- **Minor comments.** All your specific comments regarding the manuscript have been rectified or explained in the revised manuscript. Several figures have also been redrawn to resolve your concern. We hope our revisions have met your expectations.

Sincerely, we sincerely appreciate your comments to our manuscript, which pushed us to better interpret our findings. The revisions and explanations to your comments have largely improved the quality of the manuscript.

The authors

1. Detailed and accurate climate data are important for the analyses. In this study, the authors used the climate data from the CMFD dataset, which is with a coarse spatial resolution and how does this dataset consider the topography effects on climate data? Will this introduce uncertainty into your analysis?

Response:

Thanks for your comment regarding the climate dataset. We agree with you that detailed and accurate climate data is important for our study, given the importance of climate factors in regulating the temporal and spatial variations in CO₂ exchange. To resolve your concerns, we did the following work.

- During the revision, we collect the **observed** climate data for each eddy covariance tower from literatures, i.e., usually mean annual temperature or precipitation, and then compared them with the **CMFD** dataset (0.1° in spatial resolution). It turns out that the CMFD is highly correlated with the observed climate data ($R^2=0.94$ for air temperature; $R^2=0.93$ for precipitation; **Fig. R4a-b**). This suggests that the retrieved climate information from the CMFD dataset is good enough to represent the climate background for the eddy covariance tower during analyses and random forest training.
- Still, we are wondering whether a finer climate dataset would better represent local climate information. To verify above issue, we then used the highest climate dataset public available (0.0083° in spatial resolution), the **WorldClim** dataset, i.e., it is 120 times better than the CMFD in spatial resolution. Indeed, a finer climate dataset would better represent observed the climate information of the eddy covariance towers ($R^2=0.95$ for air temperature; $R^2=0.95$ for precipitation; **Fig. R4c-d**), though there is no significant improvement compared with the CMFD dataset.
- In addition, it is notable that the CMFD dataset even performs better than the WorldClim dataset, given the CMFD estimation is closer to the 1:1 line (**CMFD**: $y=0.98x$ for temperature, $y=0.95x$ for precipitation; **WorldClim**: $y=0.95x$ for temperature, $y=0.90x$ for precipitation). Based upon the facts, we consider the performance of CMFD and WorldClim are comparable, and they both could

represent the observed climate background of the eddy covariance towers.

- We expect the observed climate background of the eddy covariance towers would improve the contribution in explaining the elevation dependent pattern of NEP. In the revised manuscript, we then used the ground-based temperature and precipitation in SEM and correlation analyses. Indeed, it turned out that the observed climate dataset did perform better than before (**Fig. R2** in response letter, or **Fig. 2a** in main text).

Fig. R4. Comparison among temperature/precipitation from ground-based observations, CMFD, WorldClim, CMIP6 datasets and MsTMIP climate datasets. (a-b) Correlation between ground-based observations and CMFD; **(c-d)** Correlation between ground-based observations and WorldClim; **(e-f)** Correlation between ground-based observations and CMIP6; **(g-h)** Correlation between ground-based observations and MsTMIP forcing data (CRUNCEP). CMFD=China Meteorological Forcing Data; WC=WorldClim; MsTMIP=Multi-Scale Synthesis and Terrestrial Model Intercomparison Project; CMIP6= Coupled Model Intercomparison Project Phase 6; TAS=Air temperature; PREC=Precipitation.

- During the random forest training, we need to obtain the time series dataset for each climate factors, for each day and for each eddy covariance tower. However, most studies did not provide

complete climate information in the literature. That is why we used the CMFD dataset to build a complete dataset for NEP variation at daily scale (like temperature, precipitation). By contrast, the WorldClim dataset only covers the period of 1970-2000, while the range of the EddyChina2023 is 2002-2022, i.e., the WorldClim cannot be used in random forest training.

- Finally, we admit that the lack of high-quality ground-based observed climate dataset may have induced uncertainty to the random forest training, though we have no choice, before more detailed, high-quality datasets were released by our colleagues in China. The comparison between observed climate factors CMFD and similar datasets were then added to the “Materials and methods” as **Fig. S15**. The uncertainty arising from climate dataset is also mentioned in the “Limitations”, which could be seen in **Line 229 to 235**.

2. Another issue is regarding the comparison between eddy covariance data and MsTMIP simulations. The spatial resolution is 0.5 degree for MsTMIP models, which does not match the detailed observations of EddyChina2023. Again, the CMIP6 model projections are with even much coarser resolution for the future climate change, so what is the uncertainty for comparing the future climate change on high elevation and low elevation?

Response:

Thanks for your concern regarding the MsTMIP dataset. Indeed, we agree with you that both the MsTMIP and CMIP6 have relatively coarse spatial resolution, which may induce uncertainty due to topographic effect in mountain areas. To resolve your concern, we conduct additional work and list it here.

- Regarding the CMIP6, we also compare the outputs of temperature and precipitation from the CMIP6 models with the ground-based observations. The comparison shows that the CMIP6 ensemble agrees well with the ground-based observations (**Figure R3 e-f**), which is very much like the performance of CMFD and WorldClim, ensuring them to be used as forcing data for the random forest predictions in the future. In addition, over 20 models for each scenario are used across 4 climate scenarios in the current study, which may also provide a strong constrain to future variation in NEP.
- Regarding the MsTMIP ensemble, we also compare the ground-based observations of temperature and precipitation of each eddy covariance tower with the forcing data of MsTMIP ensemble (CRUNCEP climate forcing data). The MsTMIP ensemble generally captured the varying spatial pattern of temperature and precipitation across China (**Figure R4 g-h**). Also, the **Figure 4a** (in main text) show similar elevation dependent pattern of GPP, which is consistent to the result from ground-based GPP of the EddyChina2023.
- Sincerely, thanks a lot for pushing us to think deeper. Your comments did help us improve the quality of the manuscript.

3. One main finding of this study is that ecosystems on high mountains have higher temperature

sensitivity and future climate change will increase the CO₂ uptake in these ecosystems. However, I found weak relationship between either GPP or RE and air temperature from the main figure 2, which is the most important figure illustrating the relationship between different components of NEP and multiple factors. There seems to be contradictory between them.

Response:

Thanks for your comments.

- We employed the Pearson correlation (**Figure R5**, in supporting materials) and SEM (Figure 2a, in main text) analyses to explain the **spatial pattern** of NEP, rather than its temporal variation. The Pearson correlation analyses did find significant correlations between NEP, GPP and RE with temperature variations, including air temperature and land surface temperature. In the updated SEM analyses (**Fig. 2a**), we used the observed climate factors for each eddy covariance tower. It turned out the ground-based observations perform better than previous version, showing a clear relationship between GPP and air temperature.
- The analyses regarding the temperature sensitivity focused on the **temporal variation** of CO₂ exchange. Temperature is widely witnessed as an important factor in controlling the temporal variation of CO₂ exchange. In addition, the higher temperature sensitives, and future increase in NEP in the current study both supports the role of temperature on regulating the temporal variation in CO₂ exchange. Therefore, we suggest that temperature play a role in both the spatial pattern and temporal variation of CO₂ exchange.

Fig. R5. Correlation relationships between gross primary productivity (GPP), ecosystem respiration (RE) and net ecosystem productivity (NEP) and climate, soil and permafrost factors. LAT, latitude; LON, longitude; ELV, elevation; TAS, atmospheric temperature; PREC, precipitation; NR, reactive N; LST, land surface temperature; PAR, photosynthetic active radiation; SWC, soil water content; LAI, leaf area index; [CO₂], atmospheric CO₂ concentration.

Detailed comments

1. line 37-42, This part is not relevant to the topic of this manuscript.

Response: Thanks for your comment. This sentence has been removed from this paragraph.

2. Line 98-103, In which period are these results determined? Do you consider the potential difference

in the interannual variations in NEE among different sites in different ecosystems? Another point is that the occurrence of extreme climate events may be quite different

Response:

- These sentences aimed to provide a contemporary estimation of the NEP strength across China's various ecosystem types. The observations of eddy covariances range from 2002 to 2020, with a mean of 2012. Now this sentence has been revised to "*Among the terrestrial ecosystem types during 2002 to 2020 (with a mean of 2012)*". Please see **Line 100** in Text S1.
- Indeed, the net CO₂ sink usually show strong interannual variability, which can be captured by multiyear eddy covariance observations. We consider these eddy covariance observations (>200 sites and >500 site-year) as *random sampling* of the status of net CO₂ sink of each ecosystem type across China. In addition, it is notable that *118 sites (58% of the dataset) covered 2+ years' observation, while 79 sites (40%) provide covered 3+ years' observation*. This may have reduced the uncertainty due to interannual variability, especially due to climate extremes.
- The Line 98-103 intended to verify whether the net CO₂ sink is consistent to repeated soil sampling across each ecosystem. We notice that, for each ecosystem type, 10+ sites were available to cover the net CO₂ sink strength. In China, the croplands, forests, and grasslands cover 3/4 of China's land area. It is also notable that there were lots of *eddy covariance observations in croplands (40 sites and 116 site-year), forests (58 sites and 134 site-year) and grasslands (41 sites and 129 site-year)*. Therefore, multiple observations for each ecosystem type may have also largely reduced the uncertainty from interannual variability.
- The sentences have been added to the descriptions to the EddyChina2023 dataset in "Materials and methods". Please see **Line 260 to 261**. It is also notable that the "*Representativeness of the eddy covariance dataset*" has been moved to Text S1.

4. Line 106, In this study, the largest uptake of CO₂ is found in croplands, but not forests.

Response: Yes, we understand your concerns. We think that C loss via grain consumption in croplands should be considered when we compare the eddy covariance CO₂ sink with changes in soil C content. Now the sentence has been revised to "*Considering offsite C transport, the largest net CO₂ sink is by forests (note most C in croplands is consumed by humans, resulting almost a neutral CO₂ sink in croplands)*". Thanks for your comment to help us correct the error. Please see **Line 109-110 in Text S1** of Supporting Materials.

5. Line 112, This sentence is not clear. The CO₂ uptake in temperate grasslands of Inner Mongolia is neutral, indicating that the carbon gain is cancelled out by the carbon loss. How can this result be verified by soil carbon inventory? Do you mean repeated inventory data?

Response: Yes, the result is verified by repeated soil carbon inventory. We have revised the sentence to "*The NEP by the different terrestrial ecosystems is generally consistent with the variation in soil organic carbon during the last two decades, as shown by national-scale **repeated soil carbon***".

surveys". According to the suggestion to Reviewer #1, this part has been moved to Supporting Materials. Please see **Line 106-108** in Supporting Materials Text S1.

6. Figure 2, It is strange that the air temperature does not exert important effects on either GPP or RE across mountainous sites.

Response:

- Indeed, we agree with you that air temperature play a significant role in regulating **temporal variation** in GPP and RE, verified by lots of studies. We apologize that we did not emphasize that the **Figure 2a** (in main text) is intended to explore the controls of biotic and abiotic factors on the **spatial pattern** of GPP and RE. Furthermore, in a previous version of this manuscript, we used the CMFD-retrieved climate dataset, which may have caused a worse performance of air temperature in explaining the spatial pattern of CO₂ exchanges. During the revision, we extracted the ground-based climate factors for each site, and it did improve the performance of air temperature in explaining the spatial variation in GPP.
- In fact, air temperature is still quite important in controlling the spatial pattern of GPP and RE, illustrated in **Figure S5** (in supporting materials), the Pearson correlation analyses. The land surface temperature (LST) also plays a significant role in affecting the spatial pattern of GPP and RE. Temperature (air temperature and land surface temperature), water factors (precipitation and soil water content) and nutrient supply (N) all strongly affect the spatial pattern of GPP and RE (**Figure S5**).
- Based upon the Pearson correlation (in supporting materials) and the updated SEM analyses (in main text), we therefore conclude the temperature as a player in affecting the spatial pattern of carbon exchange.

7. Line 135-137, But from the SEM analysis, I did not find a close linkage between NEP and variations of temperature and precipitation and nitrogen deposition.

Response:

Thanks for your comments here.

- We apologize that we did not clarify that we tried to build its **indirect** relationship between NEP and environmental factors via GPP and RE, since the NEP represents the net balance between GPP and RE. This does not mean that there is none of a close linkage between NEP and variations in temperature, precipitation, and nitrogen deposition. The above descriptions have been added to the captions of **Figure 2a** (in main text). Please see **Line 548-550**.
- In addition, in a previous version of this manuscript, we used the CMFD-retrieved climate dataset, which may have caused a worse performance of climate factors in explaining the spatial pattern of GPP and RE. During the revision, we extracted the observed climate factors for each site, and it did improve the performance of environmental factors in explaining the spatial variation in GPP and RE, i.e., the R² have increase from 0.28 to 0.39 for GPP, and 0.22 to 0.29. Please see **Figure**

R2 (in Response Letter) or **Fig 2a** (in main text).

8. Figure S14, what do the different colored lines represent?

Response: We apologize for lack of legend in the Figure S14. We added legends in this figure by redrawing the Figure S14. Thanks for your help with correcting this error.

REVIEWER COMMENTS

Reviewer #1 (Remarks to the Author):

Using collected eddy covariance data, Wei et al. revealed the elevation related pattern of net ecosystem productivity (NEP) over Chinese terrestrial ecosystems. The geographical pattern of NEP was attributed into the difference in temperature sensitivity, indicating the high elevation regions had a higher increase in NEP in future. This work had some improvements and gave an implication of NEP spatial pattern and its future trend in China, but it still had many shortcomings, which prohibited its publication in Nature Communication.

1. Based on eddy covariance data, the authors found a negative elevation-NEP relation over Chinese terrestrial ecosystems, which resulted from the differences in the varying hydrothermal factors, nutrient supply and ecosystem types (L. 113- L.114). However, the authors also implied that temperature sensitivity and human activities would account for the negative elevation-NEP pattern. These results made readers confusing: which factor or factors affected the elevation-NEP pattern? Many works addressed the factors shaping NEP spatial patterns like mean climatic factors, the changing climatic factors. What were the relationships among those factors and temperature sensitivity as author mentioned?
2. The descriptions about temperature sensitivity calculation were confused. It seems that they calculated temperature sensitivity with all daily data over the specific region, which may cover the temporal variations and spatial variations of daily NEP. The numbers and distributions of sites used for calculating temperature sensitivity may introduce the biases in calculated values, though I did agree with them that the high elevation regions took higher temperature sensitivity. In addition, the dataset of daily NEP was not available, which further confused the understanding of temperature sensitivity calculation.
3. After calculating the temperature sensitivity, the authors validated the differences in the temperature sensitivity with values of modeled GPP and LAI. This validation had many gaps between NEP temperature sensitivity and GPP or LAI values, though they gave the changing rates of GPP and LAI. Higher sensitivity did not ensure more productive. NEP differed from GPP and LAI.
4. When predicting the future trend of NEP with machine learning and models, the variations of temperature sensitivity were not considered, which should introduce uncertainties in modeled NEP. This shortcoming did exist in many works, but it should be discussed at least.
5. Some sentences like L. 40- L. 42 seem unprofessional as this was a scientific article.

Reviewer #3 (Remarks to the Author):

The revised manuscript has well addressed my former comments. The manuscript is now much clearer. I suggest acceptance after addressing the minor comments.

Minor comments

1. Line 38, altitude? Or latitude?
2. line 106-107, for all EC sites? I did not find any support for this point.

Response Letter

Reference Number: NCOMMS-23-30115B

Title: Elevation-dependent pattern of net CO₂ uptake across China

To Reviewer #1

Using collected eddy covariance data, Wei et al. revealed the elevation related pattern of net ecosystem productivity (NEP) over Chinese terrestrial ecosystems. The geographical pattern of NEP was attributed into the difference in temperature sensitivity, indicating the high elevation regions had a higher increase in NEP in future. This work had some improvements and gave an implication of NEP spatial pattern and its future trend in China, but it still had many shortcomings, which prohibited its publication in Nature Communication.

Dear Reviewer,

Thanks very much for your efforts to review our manuscript for two rounds. We appreciate your new, thoughtful comments. All your comments have been carefully considered and revised. Here is a summary of our revisions.

- **Temperature sensitivity.** We validate our results with two new approaches, rather than using the changing rates. **(1)** We calculated each site's temperature sensitivity of NEP. The results validate that there is higher temperature sensitivity in high-elevation environments. **(2)** The MsTMIP ensemble also confirms the higher temperature sensitivity of NEP at high-elevation environments. **(3)** We consider the strong temperature constrain and less human impacts may have caused the higher temperature sensitivity in high-elevation environments.
- **Changing rates of NEP.** **(1)** Regarding the "NEP differed from GPP and LAI", we then analyzed the changing rates of NEP based upon the MsTMIP ensemble. The results of NEP are quite like that of the GPP. **(2)** Regarding the "Higher sensitivity did not ensure more productive", we separate the changing rates of NEP into a new section. **(3)** We then analyzed the changing rates of GPP of models and satellite observations, both confirming the stronger relative changes in productivity in high-elevation environments.
- **NEP projection.** All daily NEP data of EddyChina2023 have been used to train/validate Random Forest to build an implicit model between NEP and independent factors. This means that the temperature sensitivity has been considered in NEP projections. Trying to validate our finding, we then analyzed the default NEP of CMIP6 (without constrain from the EddyChina2023). The CMIP6 default NEP outputs generally support our findings that there would be a more robust relative change in high-elevation environments.
- **Other revisions.** We slightly revised the structure of the manuscript. Other necessary revisions have also been made to the main text, figures and supporting materials.

By responding each your comment and conducting more analyses, we believe that the revised manuscript has strengthened our findings substantially. We also hope the revised manuscript have meet your expectation. **Again, thanks very much for your contribution!**

Best regards,

Prof. Da Wei

1. Based on eddy covariance data, the authors found a negative elevation-NEP relation over Chinese terrestrial ecosystems, which resulted from the differences in the varying hydrothermal factors, nutrient supply, and ecosystem types (L. 113- L.114). However, the authors also implied that temperature sensitivity and human activities would account for the negative elevation-NEP pattern. These results made readers confusing: which factor or factors affected the elevation-NEP pattern? Many works addressed the factors shaping NEP spatial patterns like mean climatic factors, the changing climatic factors. What were the relationships among those factors and temperature sensitivity as author mentioned?

Response:

- **Manuscript structure.** We apologize that our manuscript and descriptions may have caused some misunderstanding to the reviewer. Here we clarify the structure of the manuscript. Now, our manuscript included the following sections. **(1) *Elevation-dependent pattern of NEP.*** Section 1 aims to recognize the elevation dependent pattern of NEP and to figure out its driving factors. **(2) *Higher temperature sensitivity in high-elevation environments.*** Section 2 aims to compare the difference between high- and lower-elevation environments regarding their temperature sensitivity. **(3) *Changing rates of productivity during the past four decades.*** Section 3 aims to explore retrospective variation in NEP along elevation gradient. **(4) *More robust changes in high-elevation environments in future.*** Section 4 aims to predict future variation in NEP.
- **Drivers of the spatial pattern and variation.** **(1) *Spatial pattern*** of NEP, our analyses suggest that climate factors, ecosystem types and reactive nitrogen level largely determines the spatial pattern of GPP, RE and thus NEP. **(2) On the temporal variations**, the NEP and photosynthesis showed a higher temperature sensitivity in high-elevation environments (3000–5000 m), compared with the lower-elevation environments (<3000 m). **(3) During the past decades**, high-elevation environments experience more rapid relative changes in productivity, due to higher temperature sensitivity and stronger increase in precipitation, compared with the lower-elevation environments. **(4) In the future**, Random Forest based upon EddyChina2023 and CMIP6 predicted a stronger relative change in NEP in high-elevation environments, whereas less change is expected at lower elevations, due to a reduced supply of nitrogen.
- Regarding the “**relationships among those factors and temperature sensitivity**”, we believe that the higher temperature in high-elevation environments is an intrinsic feature, validated by both eddy covariance datasets and process-based models, largely due to strong temperature limitation in these environments. Also, we think the varying environmental factors may also affect the NEP (like the precipitation in section 3 and reactive nitrogen in section 4), besides the temperature sensitivity.
- **To summarize**, the spatial pattern of NEP is affected by temperature, precipitation, ecosystem types and reactive nitrogen. For the decadal variation in NEP, the changing environmental factors and temperature sensitivities collectively affected the past and future variation in NEP along the elevation gradient. We also revised the structure of the manuscript, and we hope our explanations and revisions have well addressed your concerns.

2. The descriptions about temperature sensitivity calculation were confused. It seems that they calculated temperature sensitivity with all daily data over the specific region, which may cover the temporal variations and spatial variations of daily NEP. The numbers and distributions of sites used

for calculating temperature sensitivity may introduce the biases in calculated values, though I did agree with them that the high elevation regions took higher temperature sensitivity. In addition, the dataset of daily NEP was not available, which further confused the understanding of temperature sensitivity calculation.

Response:

- Thanks for your comment regarding the calculation of temperature sensitivity. We adopted this method following a previous study in the Arctic (Natali et al., 2019, Nature Climate Change: 852-857). Indeed, we agree with you that the numbers and distributions of sites used for calculating temperature sensitivity may have introduced some biases. For example, high-elevation environments have less observation sites, largely because they receive less research attentions and lack of traffic accessibility. The EddyChina2023 dataset is not a well-designed network, and this would make the high-elevation environments under-represented.
- Trying to resolve your concern, additional work has been done to validate our previous findings in Figure 3. We then test whether the high-elevation environments have higher temperature sensitivities, by calculating each site's temperature sensitivity of NEP and then conducting statistical analyses. The results further validate there is significant difference between high-elevation and lower elevation environments (Figure S7. This is a new figure). Please see **Line 122** in the main text.
- Regarding the data availability of daily NEP. We already uploaded the whole dataset to the submission system as Data Availability. The daily NEP dataset was also uploaded in FigureShare. A link to the FigureShare is provided here (<https://figshare.com/s/e1f7f9c13e547e422a71>), which would facilitate your review.

Fig. S7. Comparison of the temperature sensitivities of net ecosystem productivity between high-elevation environments and their lower elevation counterparts. High-elevation environments are defined as >3000 meter in elevation, while lower elevation environments as <3000 meters. The NEP sensitivity to temperature is defined as slope between NEP and temperature variation for each site ($Slope_{NEP/TAS}$; NEP, net ecosystem productivity; TAS=Atmospheric Temperature).

Each site's temperature sensitivity was independently calculated. Then, the difference of temperature sensitivities between two groups, i.e., high- and lower elevation environments, were tested with Independent-Samples T test. The mean temperature sensitivity for high- and lower-elevation environments are 0.08 ± 0.01 ($n=33$ sites) and 0.07 ± 0.01 g C °C⁻¹ ($n=78$ sites). The “**” indicates significant difference between two groups ($P < 0.05$).

Reference:

Natali, S. M. et al. Large loss of CO₂ in winter observed across the northern permafrost region. *Nature Climate Change* 9, 852-857 (2019). <https://doi.org/10.1038/s41558-019-0592-8>

3. After calculating the temperature sensitivity, the authors validated the differences in the temperature sensitivity with values of modeled GPP and LAI. This validation had many gaps between NEP temperature sensitivity and GPP or LAI values, though they gave the changing rates of GPP and LAI. Higher sensitivity did not ensure more productive. NEP differed from GPP and LAI.

Response:

- Thanks very much for your comment. When we analyzed the EddyChina2023 data and noticed that there is higher temperature sensitivity of NEP in high-elevation environments, we expected that this may benefit productivity in these environments. That is why we then analyzed the variation in NEP during the past decades. Indeed, we agree with you that higher temperature sensitivity would not ensure higher productivity. To resolve your concern, we conducted the following work.
- **More validation of temperature sensitivity.** After the calculation of temperature sensitivity for each site with a new approach (Fig. S7, This is a new figure), we then explored the temperature sensitivity of NEP from the MSTMIP ensemble. It turns out that the MSTMIP ensemble also validated our finding that the NEP at high-elevation environments is more sensitive to temperature variation than that in lower elevation environments (Fig. S8. This is a new figure). The result of Fig. S8 is consistent to both Fig. 3 and Fig. S7, therefore further validating the result based upon EddyChina2023 dataset. The above additional work has been added to the main text. Please see **Line 129-131**.
- **Analyses of changing rates of NEP and GPP.** In this part, we conducted the following work. **(1)** It is true that “NEP differed from GPP and LAI.” Therefore, during the revision, we analyzed the changing rates of NEP based upon the MSTMIP ensemble, besides the GPP. The results of NEP (Fig. 4a. This is a new figure), including averaged and changing rates of NEP across various elevation bands, are quite similar to the results based upon GPP (Fig. 4b. This is a redrawn figure), i.e., both show higher relative changing rates in high-elevation environments. **(2)** We also analyzed the changing rates of GPP based upon the GLASS product, given satellite observations provide another independent evidence. The GLASS GPP show very similar results to the findings of MSTMIP GPP and NEP (Fig. 4c. This is a new figure). Based upon the above evidences (i.e., MSTMIP NEP, MSTMIP GPP and GLASS GPP), we therefore conclude that higher-elevation environments experience more rapid relative changes in productivity.
- **Reasons to include GPP.** Besides the NEP, we also keep the analyses regarding the GPP, largely based upon the following reasons. **(1)** Satellite-based GPP observations can provide a direct and independent observation of ecosystem productivity, which would further validate the results of process-based models. **(2)** Satellite-based observations can hardly estimate NEP, while GPP

estimation is easier, like the GLASS GPP product used in the current study. **(3)** The spatial pattern and changing rates of GPP and NEP are quite similar to each other (Fig. 4a and 4b), which means that the GPP can be employed as a “proxy” to NEP. **(4)** The spatial pattern and temporal variation of NEP is better correlated with GPP, compared with the RE (Fig. 2a). In Fig. 3a-c, it also shows that the temperature sensitivity of GPP of high-elevation environments is 5 times of that in the lower elevations, highlighting the role of GPP in dominating the temporal variation of NEP along elevation gradient. Therefore, by employing the GPP, especially satellite-based GPP observations, we believe it could better understand the variation in NEP along the elevation gradient.

- Potential causes for the changing rates of productivity.** We agree with you that “Higher sensitivity did not ensure more productive”. During the revision, the changing rates of NEP or GPP was not used to validate the difference in temperature sensitivities. Instead, we separate the changing rates of NEP and GPP into a new section, named “*Changing rates of productivity during the past four decades.*”. We also added a new figure (Fig. S9. This is a new figure), trying to answer why the high-elevation environments experience more rapid changes. We examine the changes in atmospheric temperature, precipitation, reactive nitrogen, atmospheric CO₂ concentration and temperature sensitivity along elevation gradient (atmospheric CO₂ concentration map not available). We consider stronger changes in precipitation and higher temperature sensitivity in high-elevation environments may have contributed to its stronger relative increase in NEP, while the opposite is true for the lower elevation environments. These revisions have been added to the main text. Please see **Line 155-167**.
- Finally, we thank you very much for your comment here. By responding each your comment and conducting the above additional analyses, we largely strengthen our findings. Thanks very much for your contribution to this manuscript.**

Fig. S8 Temperature sensitivity of gross primary productivity, ecosystem respiration and net ecosystem productivity based upon the MsTMIP ensemble². GPP, gross primary productivity; RE, ecosystem respiration; NEP, net ecosystem productivity. The temperature sensitivity of GPP, RE and NEP of each pixel was calculated following the procedure described in Figure 3. Linear slopes were used for GPP and NEP, while the slope of RE were calculated after logarithm transformation. During the calculation, all pixels of each elevation band were employed to derive the slopes. Then, the results of these models were averaged to obtain the ensemble average and the standard deviation for each elevation band. The mean, median and variations (e.g., 25~75%) indicates the difference among

various models of the MsTMIP BG1 group (driven by climate change, nitrogen deposition, atmospheric CO₂ enrich and land use change). The MsTMIP BG1 group includes 8 models, i.e., BIOME-BGC, CLASS-CTEM-N, CLM4, CLM4VIC, DLEM, ISAM, TEM6 and TRIPLEX-GHG.

Fig. S9. Trend of atmospheric temperature, precipitation, and reactive nitrogen level during the past four decades. TAS, atmospheric temperature; PREC, precipitation; Nr, Reactive nitrogen. The variations of TAS and PREC were retrieved from the CMFD dataset³. The Nr is based on the Multi-scale Synthesis and Terrestrial Model Intercomparison Project (MsTMIP) product². During the calculation, all pixels of each elevation band were employed to derive the variations.

4. When predicting the future trend of NEP with machine learning and models, the variations of temperature sensitivity were not considered, which should introduce uncertainties in modeled NEP. This shortcoming did exist in many works, but it should be discussed at least.

Response:

- Thanks for your concern regarding temperature sensitivity during NEP projection. **(1)** In fact, all daily NEP data have been used to train and validate Random Forest model to build an implicit model between NEP and its dependent factors. This means that the difference of temperature sensitivity between high- and lower-elevation environments has already been considered in the Random Forest model and subsequent NEP projections. **(2)** Trying to further validate our findings,

we then analyzed the variations in NEP along elevation gradient, based upon the NEP outputs from the CMIP6 themselves (without constrain from the EddyChina2023 dataset). The CMIP6 NEP outputs also generally captured the more robust changes in high-elevation environments in future (Fig. S14. This is a new figure), though not that significant like the results based upon the fusion results of Random Forest projections (constrained by EddyChina2023). Therefore, the CMIP6 default NEP outputs also validate our findings that there would be a more robust relative changes in high-elevation environments in future, consistent to their variation during the past four decades. All these evidences make us to believe that the higher temperature in high-elevation environments is an intrinsic feature, captured by eddy covariances, biogeochemical models (the MsTMIP), earth system models (CMIP6) and Random Forest projections. Also, it could be seen that incorporation the temperature sensitivity into the Random Forest training, represented by training with EddyChina2023 dataset, better characterized the NEP variation along elevation gradient.

Fig. S14. Changes in net ecosystem productivity (ΔNEP) and reactive nitrogen (ΔNr) under various climate scenarios of CMIP6 models. The bar charts represent the relative changes by the end of the research period (2056–2065), within which the line extending from each bar represents the uncertainty in each climate scenario (± 1 SE). For ΔNr , their calculations used the following algorithm: $\Delta\text{Nr}_i = \text{Nr}_i - \text{Nr}_{\text{baseline}}$. For NEP, their relative changes were calculated by dividing the absolute changes in NEP by their average for 2020–2024 (the first five years): $\Delta\text{NEP}_i = (\text{NEP}_i - \text{NEP}_{\text{baseline}}) / \text{NEP}_{\text{baseline}}$. Here, NEP_i represents the NEP in year i , while $\text{NEP}_{\text{baseline}}$ represents the averaged NEP during 2020–2024. During the calculation, all pixels of each elevation band were employed to derive the annual cumulative NEP for each model under each climate scenario. Then, the results of these models were averaged to obtain the ensemble average ΔNEP for each elevation band under each climate scenario.

5. Some sentences like L. 40- L. 42 seem unprofessional as this was a scientific article.

Response: Thanks for your comment. We have now removed the sentence from the text.

To Reviewer #3

The revised manuscript has well addressed my former comments. The manuscript is now much clearer. I suggest acceptance after addressing the minor comments.

Dear Reviewer,

We are glad to see your suggestion of acceptance. We appreciate your time, efforts, and expertise to improve our manuscript. We add a sentence regarding your contribution in Acknowledgements.
Thanks again for your help to improve the manuscript.

Best regards,

Prof. Da Wei

Minor comments

1. Line 38, altitude? Or latitude?

Response: Thanks for your comment. The sentence has been revised to “*Similar to latitude, the elevation gradient has long been considered.....*”.

2. line 106-107, for all EC sites? I did not find any support for this point.

Response: Thanks for your correction. Now the sentence has been revised to “*Structural equation model analysis further validates the importance of temperature, precipitation and reactive N level to the spatial pattern in NEP*”. Please see **Line 107-108**.

REVIEWER COMMENTS

Reviewer #1 (Remarks to the Author):

This manuscript focusing on the elevation-NEP pattern improved some from its previous version and answered part of my concerns, while it still had some confusing points, which should be documented clearly before its publication.

First, the relationship between elevation-dependent pattern of NEP and temperature sensitivity should be clearly addressed. I agreed with you that higher temperature sensitivity appeared at high elevation regions. But the elevation pattern of NEP resulted from the varying hydrothermal factors, nutrient supply, and ecosystem types. Therefore, elevation-dependent pattern of NEP resulted in not from the difference in temperature sensitivity. You may tell the story with the order of elevation-NEP pattern and its drivers, the induced temperature sensitivity difference, the productivity change rates, and future projections.

Second, the implication that future changes in climate and human activities may affect this elevation-NEP pattern was too arbitrary. From this work, we can find that future changes may enlarge the difference in NEP among regions, while whether this change in NEP altered the elevation-NEP pattern needed further analysis.

Third, with data-driven results to project future changes may be careful as the data-driven results may have shortcomes with projections, which should be discussed at least.

Reviewer #2 (Remarks to the Author):

My previous comment was regarding the lack of explanation in the text of the nearly flat temperature sensitivity of GPP at low elevations.

The authors have not dealt much with that but I only addressed the editor on my comment.

Authors have done a bit more work trying to explain the difference in obtained temperature sensitivities with elevation /geographical location.

They estimated the equivalent temperature sensitivities from the MsTMIP ensemble and show it in Fig S8. the units of these figures are not comparable to the units shown in Fig3 of the manuscript.

I do wonder if the flat GPP Temperature sensitivity is obtained with the models as well.

Authors should change the units of Fig S8 to make them comparable to Fig 3.

Response Letter

Reference Number: NCOMMS-23-30115C

Title: Elevation-dependent pattern of net CO₂ uptake across China

To Reviewer #1

This manuscript focusing on the elevation-NEP pattern improved some from its previous version and answered part of my concerns, while it still had some confusing points, which should be documented clearly before its publication.

Dear Reviewer,

Thank you very much for your comments. As you can see, all your comments have been carefully considered during the revision, including the following aspects:

- **Structure.** We are pleased that the structure you proposed is basically consistent with the structure of the paper. Trying to make the manuscript easier for potential readers to read the paper clearly, we have simplified the titles of the four sections. These sections are now titled: "Elevation-dependent pattern of NEP and its drivers", "Temperature sensitivity difference between high- and lower elevation environments", "Productivity change rates during the past four decades", and "Projections of productivity change rates".
- **Projections.** You mentioned that "*... whether this change in NEP altered the elevation-NEP pattern needed further analysis*". We then analyzed the change in the NEP elevation pattern and determined that the slope of NEP would be slightly smoothed. However, this will not reverse the elevation-dependent pattern of NEP. Similar phenomena occur with temperature and phenology along elevations. For instance, even though temperatures are rising more rapidly at higher elevations, this will not reverse the negative elevation-dependent pattern of temperature.
- **Uncertainty.** Predictions based on observed data can effectively constrain the future changes of NEP, but machine learning does have a lot of uncertainty. In our case, data-based predictions may overlook underlying mechanisms. Predictions focus more on aboveground factors, while neglecting drastic changes in underground processes. Predictions rely on the availability of data, which is not always evenly distributed across different environments. These can indeed bring uncertainty to predictions. We have specifically added a section to discuss its uncertainty.

Finally, thank you very much for providing a lot of value comments during the three rounds of review. Your comments greatly changed the structure of the manuscript. At the same time, your comments also prompted us to conduct a more in-depth analysis of the results, which greatly strengthened the conclusion of this article. We greatly appreciate your time, expertise, and suggestions!

Best regards,

The authors

First, the relationship between elevation-dependent pattern of NEP and temperature sensitivity should be clearly addressed. I agreed with you that higher temperature sensitivity appeared at high elevation regions. But the elevation pattern of NEP resulted from the varying hydrothermal factors, nutrient supply, and ecosystem types. Therefore, elevation-dependent pattern of NEP resulted in not from the difference in temperature sensitivity. You may tell the story with the order of elevation-NEP pattern and its drivers, the induced temperature sensitivity difference, the productivity change rates, and future projections.

Response: We are pleased with your consistent agreement regarding the concept of higher temperature sensitivity in high elevation regions. In response to your concern, we have made structural revisions to the paper, and we are glad to know that the revised structure meets your expectations. Furthermore, we have modified the titles of four sections in the paper to enhance its overall readability. These sections are now titled: "Elevation-dependent pattern of NEP and its drivers", "Temperature sensitivity difference between high- and lower elevation environments", "Productivity change rates during the past four decades", and "Projections of productivity change rates". We appreciate the your suggestions, which have significantly improved the clarity and reader-friendliness of the results at every stage.

Second, the implication that future changes in climate and human activities may affect this elevation-NEP pattern was too arbitrary. From this work, we can find that future changes may enlarge the difference in NEP among regions, while whether this change in NEP altered the elevation-NEP pattern needed further analysis.

Response: Thank you very much for your comment here. Trying to resolve your concerns, we have conducted the following work:

- We have toned down the conclusion in the abstract: "*We therefore conclude an elevation-dependent pattern of the NEP of terrestrial ecosystems in China and highlight that future changes in Earth's climate and human activities may slightly affect this pattern, although there is significant uncertainty involved*". Please see **Line 31-33**.
- We then analyzed the elevation pattern of NEP and its changes under various climate scenarios. Our results confirm that the more rapid changes in productivity in higher elevation environments may slightly smooth the elevation-dependent pattern of NEP in the future, although there is still a large amount of uncertainty that exists. We have included the new analysis in the main text (**Line 185-188**) and have also added it as **Fig. S16** in the supplementary materials.
- The phenomenon of changing NEP slope along elevation is similar to the more rapid temperature change along elevation, known as elevation dependent warming. Another similar phenomenon is the changing phenology along elevation, meaning that higher elevation environments experience stronger greening advancement than lower elevation environments. Despite more rapid changes in temperature, phenology, and NEP in higher elevation environments (i.e. smoothing slopes of temperature, phenology, and NEP), the overall negative pattern of temperature, phenology, and NEP along elevation would never be reversed.

Fig. R1. Slope of net ecosystem productivity (NEP) along elevation and its changes under various climate scenarios constrained by the CMIP6 models and the EddyChina2023 dataset. (a) SSP1-2.6, (b) SSP2-4.5, (c) SSP3-7.0 and (d) SSP5-8.5. For the left part of each figure, the NEP slope (SL_{NEP}) represents the slope between NEP with elevation. The bar represents the 1.5 times interquartile range. For the right part of each figure, the NEP slope change was the difference between 2020–2024 and 2056–2065: $SL_{NEP} \text{ Change} = SL_{NEP-2020-2024} - SL_{NEP-2056-2065}$.

Third, with data-driven results to project future changes may be careful as the data-driven results may have shortcomings with projections, which should be discussed at least.

Response: We agree that the data-driven projections may contain some uncertainty. **(1)** The input highlights the use of statistical or implicit relationships to achieve data-driven projections between NEP and environmental factors, but it may have overlooked underlying mechanisms. Nonlinear changes and interactive effects from CO_2 concentration, climate extremes, species composition, and human management could also be challenging to capture through data-driven projections. **(2)**

Moreover, data-driven projections tend to focus more on aboveground factors such as climate and vegetation, while neglecting significant changes in underground processes, particularly in microbial processes, which may not be accurately reflected in eddy covariance observation or data-driven projections. **(3)** In addition, projections based on data rely heavily on the availability of data, which is not always evenly distributed across different environments. For example, there may be more observations in lower-elevation environments. The above discussions have been added to the main text. Please see Line **202-213**.

To Reviewer #2

My previous comment was regarding the lack of explanation in the text of the nearly flat temperature sensitivity of GPP at low elevations. The authors have not dealt much with that but I only addressed the editor on my comment.

Dear Reviewer,

Thank you for your interest in GPP's temperature sensitivity in lower-elevation environments. We would like to provide an explanation regarding your concern. We believe that there could be five possible factors that influence GPP, which include temperature, radiation, moisture, nutrients, and human management.

- **Temperature.** Strong temperature limitations may be the main reason for the difference in temperature sensitivity between high- and lower-elevation environments. High-elevation environments are 11°C colder than ecosystems below 3000 m. However, the temperature limitation to GPP is not as significant in lower-elevation environments.
- **Radiation.** Solar radiation is abundant in high-elevation areas due to the thinner air, while it is relatively weak in low-altitude areas, as noted by Tang et al. (2017). Meanwhile, air pollution is more severe in low-elevation areas, resulting in a higher concentration of aerosols that significantly affect radiation, as reported by Wang et al. (2022). The role of solar radiation in influencing GPP in lower-elevation environments cannot be ignored.
- **Nutrient or water limitation.** Lower-elevation areas usually have more nutrient and water supply, under irrigation, fertilization, and nitrogen deposition. Therefore, it is possible that nutrient and water limitations may not be the primary factors affecting the lower GPP-temperature sensitivities in lower-elevation environments.
- **Management.** Human management is influential in low-elevation areas, which can result in a lower apparent temperature sensitivity of GPP, as illustrated in the comparison between Zoige Station (at an elevation of 3500 m) and Gucheng Station (at an elevation of 15 m). At Zoige meadow, GPP closely follows the temperature variation, resulting in a higher temperature sensitivity of GPP (Figure R2 a-b). In contrast, at Gucheng Station, there is a significant decrease in GPP during the summer (wheat-corn rotation period), which leads to a relatively low apparent temperature sensitivity of GPP (Figure R2 c-d). In addition to the above phenomenon, human activities such as plantation, harvesting, logging, and land use changes in the lower-elevation environments may also contribute to the lower *apparent* temperature sensitivity of GPP.
- Therefore, we consider that temperature, radiation, and management may be responsible for the lower temperature sensitivity of GPP in lower-elevation environments. However, it must be

acknowledged that there are still knowledge gaps in this study. In fact, research regarding the variation of CO₂ exchange along the elevation gradient can be quite challenging, especially at a daily scale, because it is difficult to build eddy covariance stations in mountainous areas. In this study, we adopted a unique method to tackle the above issue and did obtain some new findings. Given the fact that there are still knowledge gaps regarding the CO₂ exchange along elevation, such as the lower temperature sensitivity of GPP, our team will continue to conduct more observations to address the above issue. The above discussions have been added to the main text. Please see Line 132 to 146.

Finally, we would like to express our sincere appreciation to you for providing us with lots of valuable comments during the three rounds of review. This greatly strengthened our conclusions and improved the quality of the manuscript. Personally, I also learn a lot from your comments. Thank you very much!

Best regards,

The authors

The following is a figure to illustrate the human disturbance to GPP-temperature relationship.

Fig. R2. Variations in temperature and GPP in natural ecosystems and human-managed ecosystems. (a-b) alpine meadow in Zoige; (c-d) winter wheat-summer corn in Gucheng. TAS refers to air temperature, and GPP refers to gross primary productivity. DOY, Day of Year. Datasource: <https://doi.org/10.57760/sciencedb.o00119.00052>; <https://doi.org/10.57760/sciencedb.o00119.00071>.

Reference: (1) Tang WJ, Qin K, Yang X et al. (2017) An efficient algorithm for calculating photosynthetically active radiation with MODIS products, *Remote Sens. Environ.*, 194, 146-154. (2) Wang Z, Wang C, Wang X. et al. (2022) Aerosol pollution alters the diurnal dynamics of sun and

shade leaf photosynthesis through different mechanisms. *Plant, Cell & Environment*, 45(10), 2943-2953.

Authors have done a bit more work trying to explain the difference in obtained temperature sensitivities with elevation /geographical location. They estimated the equivalent temperature sensitivities from the MsTMIP ensemble and show it in Fig S8. the units of these figures are not comparable to the units shown in Fig3 of the manuscript. I do wonder if the flat GPP Temperature sensitivity is obtained with the models as well. Authors should change the units of Fig S8 to make them comparable to Fig 3.

Response:

- We apologize for using different units in Fig. 3 and Fig. S8, where they were respectively represented as $\text{g C m}^{-2} \text{ day}^{-1} \text{ }^{\circ}\text{C}^{-1}$ and $\text{g C m}^{-2} \text{ yr}^{-1} \text{ }^{\circ}\text{C}^{-1}$. Following your suggestion, we have modified the units in Fig. S8. Please note that the values in Fig. S8 are equivalent to the slope in Fig. 3. For instance, the slope at high and low elevations in Fig. 3 are 0.090 and 0.016 $\text{g C m}^{-2} \text{ day}^{-1} \text{ }^{\circ}\text{C}^{-1}$ for GPP, 0.07 and 0.05 $\text{g C m}^{-2} \text{ day}^{-1} \text{ }^{\circ}\text{C}^{-1}$ for NEP, respectively. The temperature sensitivity of RE is 0.070 and 0.048 (unitless) at high and low elevations, respectively.
- The ground-based temperature sensitivity of GPP, RE and NEP are generally higher than the results based upon MsTMIP models, though they both supported generally higher temperature sensitivity at higher elevation environments.
- Regarding your comments “I do wonder if the flat GPP Temperature sensitivity is obtained with the models as well”, the results of MsTMIP models shows that the slope of GPP in low-altitude areas is indeed lower than in high-altitude areas (**Fig. S8**), which is consistent to Figure 3.

Fig. R3 Temperature sensitivity of gross primary productivity, ecosystem respiration and net ecosystem productivity based upon the MsTMIP ensemble². High, high-elevation environments; Lower, lower elevation environments; GPP, gross primary productivity; RE, ecosystem respiration; NEP, net ecosystem productivity. The temperature sensitivity of GPP, RE and NEP of each pixel was calculated following the procedure described in Figure 3. Linear slopes were used for GPP and NEP, while the slope of RE were calculated after logarithm transformation. The mean, median and variations (e.g., 25~75%) indicates the difference among various models of the MsTMIP models.